# Migraine: Advances in the Pathogenesis and Treatment

**Horia Pleș** [1], **Ioan-Alexandru Florian** [2,*], **Teodora-Larisa Timis** [3,*], **Razvan-Adrian Covache-Busuioc** [4], **Luca-Andrei Glavan** [4], **David-Ioan Dumitrascu** [4], **Andrei Adrian Popa** [4], **Andrei Bordeianu** [4] and **Alexandru Vlad Ciurea** [4]

[1] Department of Neurosurgery, Centre for Cognitive Research in Neuropsychiatric Pathology (NeuroPsy-Cog), "Victor Babeș" University of Medicine and Pharmacy, 300041 Timișoara, Romania; ples.horia@umft.ro
[2] Department of Neurosciences, "Iuliu Hatieganu" University of Medicine and Pharmacy, 400012 Cluj-Napoca, Romania
[3] Department of Physiology, "Iuliu Hatieganu" University of Medicine and Pharmacy, 400012 Cluj-Napoca, Romania
[4] Neurosurgery Department, "Carol Davila" University of Medicine and Pharmacy, 020021 București, Romania; razvan-adrian.covache-busuioc0720@stud.umfcd.ro (R.-A.C.-B.); luca-andrei.glavan0720@stud.umfcd.ro (L.-A.G.); david-ioan.dumitrascu0720@stud.umfcd.ro (D.-I.D.); andreiadrianpopa@stud.umfcd.ro (A.A.P.); andrei.bordeianu@stud.umfcd.ro (A.B.); prof.avciurea@gmail.com (A.V.C.)
***** Correspondence: florian.ioan.alexandru@gmail.com (I.-A.F.); doratimis@gmail.com (T.-L.T.)

**Abstract:** This article presents a comprehensive review on migraine, a prevalent neurological disorder characterized by chronic headaches, by focusing on their pathogenesis and treatment advances. By examining molecular markers and leveraging imaging techniques, the research identifies key mechanisms and triggers in migraine pathology, thereby improving our understanding of its pathophysiology. Special emphasis is given to the role of calcitonin gene-related peptide (CGRP) in migraine development. CGRP not only contributes to symptoms but also represents a promising therapeutic target, with inhibitors showing effectiveness in migraine management. The article further explores traditional medical treatments, scrutinizing the mechanisms, benefits, and limitations of commonly prescribed medications. This provides a segue into an analysis of emerging therapeutic strategies and their potential to enhance migraine management. Finally, the paper delves into neuromodulation as an innovative treatment modality. Clinical studies indicating its effectiveness in migraine management are reviewed, and the advantages and limitations of this technique are discussed. In summary, the article aims to enhance the understanding of migraine pathogenesis and present novel therapeutic possibilities that could revolutionize patient care.

**Keywords:** migraine pathogenesis; molecular markers; calcitonin gene-related peptide (CGRP); migraine treatment; neuromodulation

## 1. Introduction

An overwhelming majority of the global population, approximately 95%, have suffered from a headache at some point in their lives, with an alarming annual prevalence that suggests nearly half of all adults have experienced a headache within a given year [1]. The ramifications of this health issue extend far beyond personal discomfort, with headaches accounting for one-tenth of consultations with general practitioners [2], and a significant portion, one-third, of referrals to neurologists [3]. Moreover, acute medical admissions related to headaches are alarmingly high, constituting one in every five cases.

The World Health Organization recognizes the debilitating nature of headaches, including them among the top ten global causes of disability. Interestingly, in women, the prevalence and impact of headaches are even more pronounced, ranking among the top five causes of disability [4]. It is pertinent to note that the debilitating impact of headaches

is comparable to chronic conditions, such as arthritis and diabetes, and its severity exceeds that of conditions like asthma [5,6].

Taking the United Kingdom as an illustrative example, the socioeconomic implications of migraine, a specific type of headache, are vast. An estimated 25 million workdays are lost annually due to migraine, creating an indirect economic burden of nearly GBP 2 billion per year. This figure does not even include the direct healthcare costs associated with managing headaches, such as medication expenses, consultations with general practitioners, referrals to specialists, and visits to emergency care facilities [7].

Quantifying the influence of headaches, particularly migraine, on an individual's quality of life can be challenging. Yet, it is clear from reported data that the impact is substantial. A significant percentage, approximately 75% of patients, experience functional disability during a migraine attack. Additionally, half of the sufferers require the assistance of family members or friends during an attack, causing a significant disruption to their social lives [8]. The ripple effect of headaches extends beyond the individuals suffering from them, impacting society at large and requiring serious attention for more effective management strategies [9].

Migraine, a long-term headache disorder punctuated by episodic bouts, is characterized by repeated instances of severe headaches that present with unique associated symptoms. These include photophobia, a heightened sensitivity to light, and phonophobia, an increased sensitivity to sound [10]. The classification of episodic migraine—an intermittent but recurring form of this disorder—hinges on the frequency with which a patient experiences these debilitating headaches.

In the majority of cases, patients undergo fewer than 15 episodes of headaches per month, a condition identified as episodic migraine. Conversely, there is a subset of individuals who face a more frequent occurrence of headaches—on 15 or more days each month, spanning over three months. Importantly, at least eight of these days should either meet the diagnostic criteria for migraine without the accompanying aura or show responsiveness to treatment specifically designed for migraine. The International Headache Society recognizes this latter classification as chronic migraine [11].

Chronic migraine, although less common when compared to its episodic counterpart, remains a pervasive and incapacitating issue [12]. It poses a significant burden on those afflicted with the condition, dramatically impacting their daily lives and well-being [13,14]. This persistent form of migraine continues to be a widespread challenge, necessitating ongoing research and improved therapeutic strategies to ease the strain it puts on sufferers [15].

### 1.1. Brief Overview of Migraine as a Prevalent Neurological Condition

Over the past three decades, there has been a significant upsurge in the worldwide prevalence of migraine. As highlighted by the Global Burden of Disease (GBD) 2019 study, the estimated global occurrence of migraine escalated from 721.9 million (with a 95% uncertainty interval (UI) of 624.9–833.4) in 1990 to a staggering 1.1 billion (95% UI: 0.98–1.3) in 2019. The percentual shifts in global age-standardized prevalence rate and years lived with disability (YLDs) over these nearly three decades were recorded at 1.7 (95% UI: 0.7–2.8) and 1.5 (95% UI: −4.4 to 3.3), respectively [16].

In this time frame, the sharpest escalations in the age-standardized prevalence per 100,000 individuals were recorded in East Asia with a 7.9% increase (95% UI: 4.3–12%), and in Andean Latin America with an increase of 6.7% (95% UI: 2.1– 11.9%). Conversely, the most significant decreases were seen in high-income North America [−2.2% (95% UI: −5.3 to 1.1%)] and Southeast Asia [−2.2% (95% UI: −3 to −1.4%)]. Moreover, the age-standardized YLD rate due to migraine also saw an increase from 517.6 (95% UI: 82.0–1169.1) in 1990 to 525.5 (95% UI: 78.8–1194.0) in 2019.

The incidence of migraine consistently appeared higher in females than in males across all age groups. In 2019, the global age-standardized prevalence rate for females was 17,902.5 (95% UI: 15,588.3, 20,531.7) per 100,000 populations, in comparison to 10,337.6 (95% UI: 8948.0, 12,013.0) for males [16].

Notably, the most frequent incidence of migraine, both in terms of rate and absolute number of new cases, was seen in the age bracket of 10–14 years for both genders. Over the course of 2019, the number of YLDs due to migraine began to increase from birth, reaching a peak in the 30–34 age group, after which it slowly receded for both sexes [17].

Contrary to expectations, socioeconomic status did not appear to have a direct correlation with the burden of migraine. The study did not reveal any discernible link between the socio-demographic index (SDI) and the YLD rate associated with migraine. This lack of association suggests that migraine does not discriminate based on socioeconomic status, further underlining the pervasive nature of this debilitating condition [17].

The prevalence of migraine has been reported to fluctuate between 2.6% and 21.7%, with an estimated average prevalence close to 12%. However, these figures vary significantly across different nations and even between individual studies conducted within the same country [18–21].

Notably, there appears to be a strong familial connection among individuals suffering from migraine, suggesting that genetic factors significantly contribute to the risk of developing this condition [20,22–25]. Supporting this theory, twin studies have indicated that migraine represent a complex genetic disease that involves an intricate interplay between genetic and environmental factors. Remarkably, the heritability of migraine has been estimated to be as high as 65% [26–29].

However, in spite of the robust genetic implications suggested by these studies and several large-scale genome-wide association studies (GWAS) conducted over the years, the scientific community has yet to conclusively identify specific candidate genes responsible for migraine. A recent systematic re-evaluation of 27 proposed candidate genes found none to be statistically significant [30].

Interestingly, the prevalence of migraine among neurologists is significantly higher when compared to the general population, reaching prevalence rates as high as 48.6% in some studies. This elevated prevalence is most likely attributable to enhanced self-recognition of migraine symptoms among professionals who are extensively trained and experienced in diagnosing and treating the condition. This assertion is supported by a study revealing that just over half of individuals who were diagnosed with migraine actually recognized their headache as a migraine [31].

*1.2. Prevalence of Migraine in Pediatric Patients*

From a meta-analysis, in which data were sourced from 40 studies encompassing a sample of 15,626 pediatric and adolescent individuals diagnosed with migraines, an 11% prevalence rate was noted, displaying considerable heterogeneity. Among these, 27 studies delineated migraine prevalence based on gender. The aggregated prevalence rate for females stood at 4%, whereas for males it was 3%. Specific data concerning MwoA (migraine without aura) and MwA (migraine with aura) were gleaned from 13 studies, which covered 3481 and 1322 subjects diagnosed with MwoA and MwA, respectively. Prevalence for MwoA was identified at 8% and for MwA at 3%, with marked heterogeneity for both. Only six studies offered data on chronic migraines, revealing a prevalence that fluctuated between 0.2% and 12% [32].

From a separate dataset of 31 studies, information was extracted involving 13,105 pediatric and adolescent subjects diagnosed with TTH (tension-type headache). This cohort exhibited a prevalence of 17%, with notable heterogeneity. Out of these studies, 23 offered a gender-based breakdown of TTH prevalence, yielding a consolidated prevalence rate of 11% for females and 9% for males. Limited data on episodic and chronic TTH were derived from 7 studies, which presented a prevalence range of 4–29% and 0.2–12.9%, respectively [32].

Another set of data, obtained from 40 studies, encompassed 76,782 pediatric and adolescent participants diagnosed with primary headaches in general. The overall prevalence was determined at 62%, with significant heterogeneity observed. Gender-based prevalence data for primary headaches, extracted from 29 studies, showed an aggregated prevalence rate of 38% for females and 27% for males [32].

*1.3. Medical Treatments of Migraine in Children*

Recent advancements in the pharmaceutical sector have introduced a selective 5-HT1F agonist, lasmiditan, which serves as an efficacious acute treatment for adults, demonstrating no vasoconstrictor activity. This drug is currently under investigation for its applicability in pediatric populations. Additionally, several novel calcitonin gene-related peptide (CGRP) antibodies and antagonists, which have demonstrated efficacy in both the acute treatment and prevention of migraines in adults, are now being assessed in pediatric clinical trials. In adult medical practices, there is an increasing inclination towards peripheral nerve blocks and botulinum toxin; however, the need for robust evidence supporting their efficacy in children is paramount. Furthermore, the introduction of electroceuticals—therapeutic electric devices—has broadened the treatment horizon. These devices include the external trigeminal nerve stimulator (e-TNS), non-invasive vagal nerve stimulator (nVNS), single-pulse transcranial magnetic stimulator (sTMS), and remote electrical neuromodulation device (REN). Presently, substantial evidence supporting their effectiveness in pediatric populations remains elusive; furthermore, while significant progress has been observed, it predominantly benefits the adult demographic. There is an imperative need to expedite migraine research focusing on children [33].

## 2. Pathogenesis of Migraine: Role of Molecular Markers in Identifying Migraine Triggers and Mechanisms

*2.1. Definition and Significance of Molecular Markers*

Biomarkers, in the realm of medical and biological research, are defined as quantifiable indicators of biological conditions, representing either physical manifestations or results obtained from laboratory tests that correlate with biological processes. These markers have the potential to serve critical diagnostic or prognostic functions [34]. A more explicit definition of biomarkers was proposed during a conference hosted by the US Food and Drug Administration. In this context, biomarkers are characterized as quantifiable attributes that can be objectively measured and assessed, providing insights into standard biological, pathological, or pharmacological processes [35].

This clear and precise definition paves the way for a bifurcation of biomarkers into the following two unique types: diagnostic and therapeutic. Diagnostic biomarkers serve as flags for pathological conditions and bear a close association with the risk of developing a disease and its severity. They aid in identifying the presence of a disease and gauging its stage or intensity, thus playing a crucial role in guiding clinical decision-making [36].

On the other hand, therapeutic biomarkers hold a different but equally important role. They provide information on a treatment's response, effectively serving as indicators of the efficacy or success of a therapeutic intervention. These biomarkers help clinicians tailor treatments to individual patients, allowing for personalized medicine approaches. They offer a chance to predict whether a patient is likely to respond positively to a particular treatment, making them a powerful tool in the management and treatment of diseases. By providing an early indication of the effectiveness of a therapeutic regimen, these markers can guide healthcare professionals in adjusting treatments as necessary, minimizing the trial-and-error aspect of disease management and increasing the probability of successful outcomes [15].

Biomarkers represent objective physical traits that can be harnessed to illuminate and distinguish the biological nature and mechanisms of various diseases and syndromes. Essentially, they provide snapshots of the body's physiological state and can offer valuable insights into health and disease processes. Biomarkers have an extensive range of potential manifestations, which can include but are certainly not limited to, results obtained from the examination of blood, urine, muscle, nerve, skin, or cerebrospinal fluid [37].

Additionally, biomarkers may also be identified in the form of genes or gene products. These genetic markers offer a unique insight into an individual's inherent disease susceptibility or resistance and can often illuminate potential therapeutic pathways. Likewise, biomarkers can be identified through advanced imaging techniques such as X-rays,

magnetic resonance imaging (MRI), or computed tomographic (CT) scans. These imaging biomarkers can provide a visual representation of disease progression, allowing clinicians to identify anatomical or functional changes in the body over time [34].

Another fascinating domain of biomarkers lies in the realm of electrophysiological measurements, such as those generated by electrocardiograms (ECGs), electroencephalograms (EEGs), or nerve conduction studies. These types of biomarkers record the electrical activity of the heart, brain, or nerves, respectively, offering a unique insight into the physiological function of these systems.

An important issue worth mentioning is those paraclinical investigations offer a new avenue for the management of migraine but are not proven to be of high sensibility and sensitivity for daily physician's practice. Even though neuroimaging and functional analyses of the brain activity might give a broader point of view regarding therapeutic possibilities, those should not be taken into consideration as absolute clinical criteria.

Ultimately, a biomarker could be virtually any characteristic that can be detected, quantified, and expressed in terms of physical qualities. These could include diverse measures, such as height, weight, depth, voltage, luminescence, resistance, viscosity, width, length, volume, or area. Each of these measures contributes to the vast array of biomarkers that hold promise for enhancing our understanding of diseases and guiding the development of effective therapeutic interventions. The utilization of such a wide array of biomarkers allows for a comprehensive, multi-faceted approach to understanding and treating diseases, ultimately leading to more effective and personalized healthcare solutions [38].

### 2.2. Identification of Potential Molecular Markers Associated with Migraine

The National Institutes of Health Biomarkers Definitions Working Group, in 1998, presented a definition for biomarkers. As per their definition, a biomarker refers to "a characteristic that can be objectively measured and evaluated as an indicator of normal biological processes, pathogenic processes, or pharmacological responses to a therapeutic intervention" [35]. Biomarkers may be classified based on their functional roles, such as diagnostic, therapeutic, risk, progression, and prognostic indicators.

The 'ideal' biomarker is characterized by the following features [39]:

- High sensitivity and specificity: this ensures that the biomarker can accurately identify individuals with a specific condition, and also correctly rule out those without the condition;
- High predictive value: the biomarker should be able to accurately forecast the course of the disease, providing valuable insights for disease management;
- Analytical stability: the biomarker should remain consistent over time and across different conditions, thereby ensuring reliable results;
- Easy, cost-effective, and minimally invasive analysis: the method of assessing the biomarker should be simple, economical, and cause minimal discomfort to the patient;
- Repeatability of method: the assessment method should yield consistent results when repeated, thereby ensuring the reliability of the biomarker.

In the context of migraine, however, there are no validated biomarkers due to the absence of substance or genetic variants that are exclusively associated with this condition or the lack of comprehensive studies on potential biomarkers.

#### 2.2.1. Markers of Inflammation and Oxidative Stress

The markers of inflammation and oxidative stress have been associated with migraine in several studies. Proinflammatory cytokines, such as interleukin-1 (IL-1) and interleukin-6 (IL-6), have been implicated in this condition [40]. It has been found that the level of IL-1$\alpha$ is elevated in the blood of children suffering from migraine with aura (MA) [40]. Similarly, adults with MA have been found to exhibit higher plasma levels of IL-1$\beta$ during headache-free periods and early stages of attacks as compared to those suffering from migraine without aura (MO) [40,41].

The concentration of IL-6 is reported to increase during the initial two hours of a migraine attack. Additionally, the levels of IL-10 and tumor necrosis factor alpha (TNF-$\alpha$) are also found to be elevated during these attacks. It is believed that other inflammatory markers associated with vascular dysfunction, such as homocysteine (Hcy) and matrix metalloproteinase-9 (MMP-9), are also elevated in the blood of individuals with migraine [15].

Elevated serum Hcy concentration has been linked to migraine with aura (MA), and some studies have noted a relationship between increased Hcy levels and higher frequency and severity of migraine; however, these findings are not supported by all research. Hyperhomocysteinemia (elevated Hcy) is hypothesized to initiate migraine with aura attacks through changes in pain threshold [42,43].

### 2.2.2. Markers Associated with Pain Transmission and Emotions

Biochemical research has revealed several metabolic irregularities in the synthesis of neuromodulators and neurotransmitters associated with migraine, particularly migraine without aura (MO). Alterations in the metabolic pathway of tyrosine, for example, lead to abnormal production of neurotransmitters like noradrenaline (NE) and dopamine (DA). This process results in an increase in the levels of trace amines, such as tyramine, octopamine, and synephrine. Such changes compromise mitochondrial function and elevate glutamate concentrations within the central nervous system (CNS), as can be seen in Table 1 [43].

These imbalances in the neurotransmitter and neuromodulator levels within the dopaminergic and noradrenergic synapses of pain pathways could potentially activate the trigeminovascular system (TGVS), causing the release of the calcitonin gene-related peptide (CGRP). This chain of events is believed to directly trigger migraine attacks [44,45].

CGRP plays a key role in transmitting pain signals and promoting inflammation. Its release is stimulated by the activation of TGVS and severe migraine episodes. Infusion of CGRP has been observed to provoke migraine-like attacks in patients with migraine with aura (MA). It has been reported that during inter-attack periods, the saliva and plasma levels of CGRP in migraine patients are significantly higher compared to healthy individuals [43].

Research conducted on cultured trigeminal neurons suggests that migraine treatment strategies can inhibit CGRP transcription and curtail its release, while tumor necrosis factor alpha (TNF-$\alpha$) may stimulate the transcription of this peptide [15]. Another study proposes that high levels of CGRP in saliva may correlate with a significantly improved response to rizatriptan treatment, suggesting that CGRP could serve as a valuable therapeutic marker [46].

Glutamate, which could potentially activate pathways involving both TGVS and cortical spreading depression (CSD), has been found in elevated concentrations in the plasma, platelets, and cerebrospinal fluid (CSF) of migraine sufferers, including those with chronic migraine. Research suggests that a reduction in plasma glutamate levels could be a marker of a positive response to prophylactic treatment in MO patients [43].

Serotonin (5-HT) release from platelets into the plasma may be implicated in the pathophysiology of the aura phase of migraine. Izzati-Zade observed a depletion of 5-HT stored in platelets during migraine attacks; moreover, a pattern has been observed in which the plasma level of 5-HT decreases between migraine attacks and the level of the corresponding metabolite, hydroxyindoleacetic acid (5-HIAA), increases. This pattern reverses during migraine attacks [47,48]. This correlation suggests that low 5-HT levels might enable the activation of the trigeminovascular nociceptive pathway triggered by CSD, thus supporting the hypothesis that migraines are a syndrome of low serotonergic disposition.

Additionally, a significantly higher concentration of hypocretin-1, a wakefulness-promoting neuropeptide, has been detected in the CSF of patients with chronic migraine, and this has been observed to correlate with painkiller usage [49,50]. Elevated hypocretin-1

levels may be indicative of the early stages of a migraine attack. Conversely, a study involving patients with cluster headaches reported reduced hypocretin-1 levels in the CSF, suggesting that low hypocretin-1 concentrations might reflect insufficient antinociceptive activity in the hypothalamus [51].

New therapeutic targets for migraine treatment, such as CGRP receptor antagonists, anti-CGRP antibodies, 5-HT1F agonists, glutamate antagonists, and dual hypocretin-1 receptor antagonists, are currently under investigation in phase II clinical trials [52,53]. These emerging therapies reflect the continuous exploration and evolution of our understanding of migraine pathophysiology.

**Table 1.** Molecules with altered CSF (cerebrospinal fluid) concentrations in patients with migraine.

| Molecule | Migraine Type (Chronic Migraine [CM]/Episodic Migraine [EM]) | Action in Relation to Migraine |
|---|---|---|
| **Sodium [54,55]** | EM | • During a migraine, there is an increase in cerebrospinal fluid (CSF) sodium concentration, while the blood plasma sodium concentration remains unchanged. Additionally, sodium excursions may follow a temporal pattern that worsens migraine in susceptible patients |
| **Homocysteine [56]** | EM | • High levels of homocysteine are potentially linked to migraine with aura and an increased risk of cardiovascular events in patients with migraine |
| **3,4-Dihydroxyphenylacetic acid (DOPAC) [57]** | EM | • Related with dopaminergic activity<br>• Positive correlation between the concentration of DOPAC (3,4-dihydroxyphenylacetic acid) and the intensity of migraine, whether with or without aura |
| **Phosphatidylcholine-specific phospholipase C [58]** | EM | • The process involves the hydrolysis of phosphatidylcholine, resulting in the production of important second messengers, diacylglycerol, and phosphorylcholine |
| **Transforming growth factor-β1 [59]** | EM, CM | • An anti-inflammatory cytokine |
| **Interleukin-1 receptor antagonist [59]** | EM, CM | • Proinflammatory cytokine |
| **Monocyte chemoattractant protein-1 [59]** | EM, CM | • Proinflammatory cytokine |
| **Corticotrophin-releasing factor [60]** | CM, MOH | • May be involved in activation of hypocretin/orexin system. |
| **Orexin-A (also referred to as hypocretin-1) [60]** | CM, MOH | • Involved in the maintenance and regulation of various physiological functions, including arousal, sleep, appetite, drinking behavior, central control of autonomic activity, certain endocrine responses, and pain modulation |
| **Glial cell line-derived neurotrophic factor [61]** | CM | • It may play a role in pain relief by regulating the expression of sodium channel subunits, capsaicin VR1 receptors, and substance P release<br>• Reduced levels found in patients with migraine |
| **Somatostatin [61]** | CM | • Regulatory anti-inflammatory and antinociceptive peptide |
| **Glutamate [62]** | CM | • The primary excitatory neurotransmitter in the central nervous system. It has been linked to various migraine-related processes, including cortical spreading depression, trigeminovascular activation, and central sensitization. |

**Table 1.** *Cont.*

| Molecule | Migraine Type (Chronic Migraine [CM]/Episodic Migraine [EM]) | Action in Relation to Migraine |
|---|---|---|
| **Tumor necrosis factor-α** [63] | CM | • A proinflammatory cytokine that plays a significant role in brain inflammatory and immune processes, as well as in the initiation of pain |
| **Taurine** [64] | EM, CM | • Inhibitory effect on neuronal activity and vasodilating properties |
| **Glycine** [64] | EM, CM | • Inhibitory neurotransmitter |
| **Glutamine** [64] | EM, CM | • May be involved with initiation and propagation of spreading cortical depression |
| **Neuropeptide Y** [65] | Acute migraine | • Strong vasoconstrictor |

## 3. Other Biomarkers Associated with Increased Risk for Migraine

### 3.1. Genetic Markers and Migraine

Many scientific investigations have striven to identify specific genetic mutations or polymorphisms that might contribute to an increased risk of developing migraine. However, as of now, none of these findings have been implemented in standard clinical practice. One rare subtype of migraine, known as familial hemiplegic migraine (FHM), which is characterized by aura and transient hemiplegia, has a well-understood genetic basis. There are three known genes where mutations have been linked with FHM—CACNA1A (FHM1), ATP1A2 (FHM2), and SCN1A (FHM3)—and this condition is inherited in an autosomal-dominant fashion [66].

The identified mutations connected to FHM lead to alterations in calcium and sodium channel functions, which are integral components of neuronal communication and excitability. Interestingly, these genetic variants have also been associated with other neurological disorders, including ataxia and childhood epilepsy [66]. Nevertheless, these mutations have not shown a strong correlation with common forms of migraine (with or without aura) or other types of headaches.

A recent study discerned a significant genetic correlation linking migraine risk to intracranial volume (rG = −0.11, P = $1 \times 10^{-3}$). This correlation was not observed in relation to any subcortical region. Notwithstanding, the study pinpointed concurrent genomic overlap between migraines and all brain structures. Gene enrichment in these mutual genomic regions indicated potential associations with neuronal signaling and vascular regulation. Furthermore, the research suggested a potential causative link between reduced overall brain volume, as well as the volume of the hippocampus and ventral diencephalon, and heightened migraine risk. Additionally, a causative correlation was proposed between heightened migraine risk and an expanded amygdala volume. Through the utilization of comprehensive genome-wide association studies, the study illuminated shared genetic pathways influencing both migraine risk and various brain structures. This suggests that variances in brain morphology in individuals with elevated migraine susceptibility could be rooted in genetics. Delving deeper into these findings offers support to the neurovascular premise of migraine origin, highlighting prospective therapeutic avenues [67].

Another study elucidated the following genes associated with familial hemiplegic migraine [68]:

- FHM1: CACNA1A—This gene undergoes missense mutations resulting in a gain of function, alongside rare large exonic deletions or deletions at the 5′ non-coding end promoter. It codes for the Alpha-1 subunit of the neuronal Cav2.1 (P/Q type) voltage-gated calcium channels, crucial for modulating neuronal excitability at the presynaptic end of glutamatergic synapses;

- FHM2: ATP1A2—Characterized by missense mutations, rare small deletions, or truncating mutations and frameshifts. It encodes the catalytic alpha-2 subunit of glial and neuronal ATP-dependent transmembrane Na+/K+ pumps, pivotal for extracellular K+ clearance and establishing a Na+ gradient, which is indispensable for glutamate reuptake;
- FHM3: SCN1A—Experiences missense mutations (gain of function) and is responsible for the Alpha-1 subunit of neuronal Nav1.1 voltage-gated sodium channels. It plays a key role in propelling action potentials of cortical neurons, predominantly in GABAergic inhibitory interneurons;
- FMH4: PRRT2—Noted for missense mutations, this gene codes for the pre-synaptic proline-rich transmembrane protein. It interacts with the synaptosomal-associated protein 25 (SNAP25), implying a potential role in merging synaptic vesicles with the plasma membrane.

Two FHM1 knock-in (KI) transgenic mouse models have been established as per references [69,70]. The KI model for the R192Q mutation, linked with pure FHM1, does not exhibit clinical anomalies. In contrast, the KI for the S218L mutation, attributed to severe FHM1, presents cerebellar ataxia, transient hemiparesis, and epilepsy. As outlined in [71], these FHM1-KI mice demonstrate heightened CaV2.1 currents and neurotransmitter release, an imbalance in cortical neurotransmission, amplified excitatory transmission in the visual cortex, and a higher vulnerability to cortical spreading depression (CSD).

Various models of FHM2-KI transgenic mice have been developed. Heterozygous transgenic mice [72] display no clinical changes but have an elevated predisposition to CSD. Mice with a partial knock-out (KO) of ATP1A2 also demonstrate a heightened vulnerability to CSD [73]. Another model with a complete KO of ATP1A2 in astrocytes manifests episodic paralysis and spontaneous CSD waves coupled with diminished EEG activity. Aberrations in brain metabolism were observed with increased levels of serine and glycine. Interestingly, a diet devoid of serine and glycine curtailed paralysis episodes in these mutants [74].

For FHM3, multiple SCN1A mutations have been documented, with the majority being missense alterations leading to enhanced function [75]. A mouse model harboring the L1649Q variant exhibited an increased susceptibility to CSD, attributed to Na+ channel inactivation defects and augmented Na+ currents, causing hyperactivity in inhibitory interneurons.

With respect to FHM4, mutations in PRRT2 have been discovered in numerous instances as referenced in [76]. A significant portion of these cases were pure FHM, while others exhibited accompanying epilepsy, cognitive impairments, or dyskinesia. PRRT2-KO mice displayed paroxysmal abnormal movements early in life, progressing to unusual audiogenic motor behaviors in adulthood and a reduced seizure threshold. Notably, both human and mouse homozygous KO-PRRT2 neurons in culture exhibited hyperactive NaV1.2 and NaV1.6 channels, inferring PRRT2's inhibitory effect on voltage-gated sodium channels as described in [77].

Researchers have also employed genome-wide association studies (GWAS) to pinpoint genes linked to an elevated susceptibility for migraine (see Table 2). In one such investigation, genetic information from 5122 individuals afflicted with migraine and 18,108 control participants was scrutinized. This scrutiny led to the identification of several specific genetic variations known as single-nucleotide polymorphisms (SNPs), which displayed significant connections to migraine. Noteworthy among these were rs2651899 (positioned on chromosome 1p36.32, close to the PRDM16 gene), rs10166942 (situated on 2q37.1, near TRPM8), and rs11172113 (positioned on 12q13.3, near LRP1). It is important to highlight that although rs2651899 and rs10166942 could be differentiated between migraine and non-migraine headaches, these three SNPs did not exhibit exclusivity for migraine with or without aura, nor were they tied to specific migraine characteristics. Nonetheless, the biological significance of these connections is substantiated by the established functions of TRPM8 in neuropathic pain and LRP1 in glutamatergic synaptic transmission [78].

Another GWAS pinpointed the following two susceptibility loci for migraine without aura: MEF2D and TGFBR2 [79]. It is important to bear in mind that the results from GWAS carried out have not overlapped so far, and larger-scale studies are necessary to confirm and expand the findings of smaller investigations and to permit the use of meta-analytical methodologies.

A migraine GWAS study from 2021 [80] identified 79 independent loci significantly correlated with migraine. This study was ethnically diverse, encompassing participants of East Asian, African American, and Hispanic/Latino origin, and consisted of 28,852 cases versus 525,717 controls.

The latest migraine GWAS from 2022 by Hautakangas et al. comprised 102,084 cases against 771,257 controls. This study unearthed 123 unique loci associated with migraines, 86 of which were newly discovered post the 2016 GWAS. Further studies even expanded independent SNPs to 167. The 2022 GWAS [81] underscored both vascular and CNS tissues/cell types. Newly detected loci encoded migraine drug targets, such as CGRP (CALCA/CALCB) and serotonin 1F receptor (HTR1F). Significantly, CGRP is the objective for CGRP antibodies, and HTR1F is targeted by ditans. Moreover, an in-depth assessment of roughly 30,000 patients from the 2022 GWAS with a precise migraine diagnosis revealed unique risk variants for specific migraine types.

The research presented thus far suggests that, aside from FHM, we are only at the preliminary stage of identifying genes significantly associated with migraine risk [82]. This observation is further illustrated by the inconsistent findings from studies investigating specific associations in migraine patients with and without aura (summarized in Table 2). For instance, one study found a significant association between a polymorphism in the gene encoding the dopamine D2 receptor (see Table 2) and migraine without aura [83]. Meanwhile, another study supported the association of DBH and SLC6A3 genes with migraine with aura [62]. Contradictorily, other investigations did not corroborate these associations in migraine patients, whether with or without aura. This variability is not unusual in genetic studies investigating diseases with a multifactorial etiology. As such, further research is needed to unravel the complex genetic underpinnings of migraine [15].

**Table 2.** Genetic mutations/polymorphisms associated with increased risk for migraine and relation to migraine.

| Gene Product | Migraine Type/Features | Action in Relation to Migraine |
|---|---|---|
| Dopamine type 2 (D2) receptor [23] | Migraine with and without aura [23,84] | • Vasoconstriction<br>• Reduces trigeminal nerve activation<br>• Inhibits release of vasoactive neuropeptide<br>• Interrupts pain transmission centrally |
| Glutathione S-transferase [85] | Migraine without aura [85] | • Increases susceptibility to environmental xenobiotic-induced migraine attacks in GSTM1 genotype |
| Dopamine type 4 (D4) receptor [86] | Migraine without aura [86] | • A potential genetic association exists between dopamine D4 receptor gene and migraine without aura |
| Tumor necrosis factor-α [87] | Migraine without aura [88] | • Proinflammatory cytokine |
| Methyltetrahydrofolate reductase (MTHFR) C677T allele [89] | Migraine with aura [89] | • MTHFR C677T polymorphism may increase homocysteine levels associated with migraine with aura |
| Dopamine β-hydroxylase gene [90] | Migraine with aura [90] | • An intracellular enzyme catalyzing the conversion of dopamine to noradrenaline; imbalance may increase susceptibility to migraine |
| Angiotensin-converting enzyme allele [91] | Migraine with and without aura [92–94] | • Involved in vasoconstriction and vascular remodeling |

**Table 2.** *Cont.*

| Gene Product | Migraine Type/Features | Action in Relation to Migraine |
|---|---|---|
| Hypocretin receptor 1 [95] | Migraine without aura [95] | • Neuropeptide generated within the clusters of nerve cells in the hypothalamus that could potentially play a role in feelings of tiredness, frequent yawning, heightened drowsiness, and strong urges for food linked to migraine. |
| Syntaxin 1A [96] | Migraine without aura [96] | • Involved in the control of brain chemicals, such as serotonin and gamma-aminobutyric acid (GABA). |
| Cytochrome P450 (CYP) 1A2 [97] | Chronic migraine [97] | • CYP1A2*1F is connected to excessive use of triptan medications, and among those who misuse these drugs, it also impacts the drug response. |

### 3.2. Recent Genetic Findings and Migraine

In a newly published family-based association study, significant markers connected to migraine were discovered, alongside genes believed to contribute to or modify the phenotypic expression of migraine within a substantial region of chromosome 6p12.2–p21.1. This region is recognized by the locus name MIGR3. Regrettably, due to the vastness of this area of interest, it is currently not feasible to pinpoint a singular gene; however, it is anticipated that future investigations employing more refined sequencing methodologies will eventually lead to the identification of a promising candidate gene implicated in migraine [98,99].

Despite the growing body of evidence suggesting that genetic factors play a pivotal role in the development of migraine, efforts to uncover the specific genes responsible for the common forms of migraine have only yielded modest success. As scientific collaboration expands on a global scale, the chances of identifying additional genetic variants linked to migraine are likely to increase. Furthermore, the unraveling of the genetic intricacies underlying polygenic diseases could potentially shed new light on the molecular pathways implicated in the pathophysiology of migraine. By extending our understanding of the genetic aspects of migraine, we may pave the way for the development of more effective diagnostic tools and therapeutic interventions.

### 3.3. Inflammatory Indicators and Migraine

Interleukins, specifically IL-1 and IL-6, have been linked with the occurrence of migraine. These cytokines, characterized by their proinflammatory nature, are believed to play a role in vascular dysfunction. Studies indicate that children experiencing migraine have raised plasma levels of IL-1$\alpha$ compared to those who do not suffer from migraine, and these concentrations are markedly higher in individuals with migraine accompanied by aura compared to those without aura [40]. Furthermore, adults experiencing aura migraine have significantly elevated plasma levels of IL-1$\beta$ during periods free from headaches and during the early onset of migraine attacks, in comparison to individuals with migraine that do not present with aura [41]. IL-6 levels also exhibit a surge in the initial two hours of a migraine attack when measured from blood samples taken from the jugular vein [50].

Other cytokines, including IL-10 and tumor necrosis factor alpha (TNF-$\alpha$), have shown associations with migraine. During migraine attacks, there are elevated serum levels of IL-10 and TNF-$\alpha$ [100]; moreover, between attacks, TNF-$\alpha$ levels in plasma are higher in children who suffer from migraine compared to those who do not. The connection between TNF-$\alpha$ and migraine is particularly noteworthy, given the repeated association of elevated levels of this cytokine with endothelial dysfunction [63]. While some studies propose that patients with migraine may have compromised endothelial function, others contradict these findings [15].

Further inflammatory markers, which are considered to be linked to vascular dysfunction, are found to be elevated in the blood of migraine patients. Research has demonstrated

that average plasma levels of C-reactive protein and homocysteine are higher in children who suffer from migraine compared to those who are not plagued by headaches [54]. Evidence also suggests that premenopausal women with migraine, especially those with aura, show signs of increased endothelial activation—a component of endothelial dysfunction—evidenced by elevated levels of von Willebrand factor, C-reactive protein, nitrate/nitrite, and tissue-type plasminogen activator antigen [55]. Markers linked to vascular repair and remodeling processes have also shown an association with migraine.

Investigations in both human subjects and animal models have proposed that matrix metalloproteinase-9 (MMP-9) might protect against the development and destabilization of plaques [61]. Moreover, patients experiencing migraine have been found to have significantly higher plasma levels of MMP-9 compared to healthy individuals and those with tension-type headaches. The average plasma MMP-9 levels were highest in subjects who had their blood samples taken between two and four days post their latest attack, implying that the elevated MMP-9 might be an indication of structural damage and subsequent remodeling associated with migraine attacks [64].

While the findings summarized here point to a relationship between various inflammatory mediators and migraine, additional research aimed at understanding the biological implications of these inflammatory mediators is necessary to confirm the validity of these potential indicators.

### 3.4. Contribution of Imaging Techniques in Understanding Migraine Pathology

3.4.1. Overview of Imaging Methods Used in Migraine Research

The advent of neuroimaging technologies has brought significant advancements in our understanding of migraine mechanisms and has enabled us to pinpoint secondary structural and functional impacts resulting from migraine. Imaging conducted during a migraine episode has helped the scientific community progress from a strictly vascular understanding of migraine pathophysiology, to a neurovascular theory, and currently towards a central nervous system (CNS) model.

Through the lens of neuroimaging, we have gained substantial ground towards unearthing the elusive "migraine generator"—the structure that triggers the initiation of a migraine episode. The investigative power of neuroimaging has shed light on the role of central sensitization in the pathophysiology of individual migraine attacks and in the progression of the disease. It has also enhanced our understanding of medication overuse headaches, along with the mechanisms by which abortive and prophylactic medications for migraine work [58].

One pivotal discovery has been the identification of cortical spreading depression (CSD) during migraine, with or perhaps without the accompaniment of an aura. This phenomenon is a wave of hyperactivity followed by a wave of inhibition in neuronal activities, which spreads across the cortex of the brain. It has been increasingly recognized as an important part of migraine pathophysiology [65].

Moreover, it has become evident through neuroimaging that individuals prone to migraine undergo structural and functional brain alterations in the periods between migraine attacks. These alterations appear to be correlated with both the duration of the disease and its severity, suggesting a possible link between more severe disease manifestation and persistent abnormalities between migraines. In essence, the affliction may not be limited to the episodes of the migraine attack but may present as a continuous, cyclic process with long-term impacts on brain structure and function. Neuroimaging has revolutionized the exploration and understanding of migraine, enabling us to visualize the structural and functional impacts of this condition on the brain, refine our understanding of its pathophysiology, and develop more effective therapeutic strategies. However, despite these strides, the complex nature of migraine warrants further research to fully understand the intricate interplay between genetic, environmental, and neurobiological factors in the manifestation and progression of the disease [101].

3.4.2. Findings and Insights Gained through Imaging Studies

Given the episodic and largely unpredictable nature of individual migraine attacks, conducting imaging during a spontaneous migraine has posed substantial challenges. To circumvent this issue, some researchers have induced migraine attacks in subjects by exposing them to known triggers, such as photic stimulation, physical exertion, or nitroglycerin [102]. A handful of investigators have successfully captured the onset of a migraine headache, while others have performed imaging immediately after the headache's initiation. Nevertheless, the total number of studies that have managed to image a migraine attack in progress remains relatively small.

## 4. Central Sensitization

The phenomenon of central sensitization in individuals prone to migraine leads to heightened pain perception during a migraine, a condition known as cutaneous allodynia, and may contribute to the progression from episodic to chronic migraine [102]. Approximately 65% of migraine sufferers develop cutaneous allodynia during individual headache episodes [103–105]. Patients with allodynia experience the skin becoming painfully sensitive to stimuli that are normally harmless, such as a light touch. Developing ways to block or reverse central sensitization could potentially alleviate migraine pain and lower the likelihood of episodic migraine evolving into chronic migraine.

One of the challenges in neuroimaging of central sensitization is differentiating between changes that arise from the increased pain sensation of cutaneous allodynia and those from structures that may specifically mediate the onset and maintenance of central sensitization. Recent advances in functional magnetic resonance imaging (fMRI) studies offer some progress in this regard. Utilizing the heat/capsaicin model of sensitization, fMRI studies have identified activation in the midbrain reticular formation region that appears specific to central sensitization [106,107]. Investigators propose that this activation occurs in the nucleus cuneiformis and the rostral superior colliculi/periaqueductal gray area.

Further fMRI studies investigated the influence of gabapentin, a medication commonly used to treat nerve pain, on brain activations following painful mechanical stimulation of normal skin compared to skin with capsaicin-induced secondary hyperalgesia [92]. Under both conditions, gabapentin reduced activations in the operculoinsular cortex. Interestingly, it was only in the presence of central sensitization that gabapentin was able to reduce activations in the brainstem and suppress stimulus-induced deactivations, suggesting that gabapentin might be more effective at reducing painful transmission when central sensitization is present. These insights set the foundation for additional investigations aimed at pinpointing the site where gabapentin acts to affect central sensitization. Unraveling this could provide valuable information for the development of future therapies aimed at inhibiting central sensitization, thereby offering a potential new avenue for migraine treatment [101].

*Insights into Migraine Pathology: From Current Pathophysological Understanding to Peripheral Interactions and Plasma Protein Extravasation*

A number of laboratory studies conducted during the 1990s postulated that the pain associated with migraine might arise from a sterile, neurogenically mediated inflammation of the dura mater, the thick membrane that surrounds the brain. Evidence of neurogenic plasma extravasation, the process whereby plasma proteins pass out of small blood vessels into surrounding tissues, has been observed during the electrical stimulation of the trigeminal ganglion in rat models. Interestingly, this process of extravasation can be halted by substances such as ergot alkaloids, indomethacin, acetylsalicylic acid, and the serotonin 5HT1B/1D agonist, sumatriptan [84].

Adding to this, preclinical research has suggested that a phenomenon known as cortical spreading depression, a wave of hyperactivity followed by a wave of inhibition in the brain, could act as a potent trigger for the activation of trigeminal neurons [85]. However, this notion has been a subject of controversy and ongoing debate in the scientific community.

Notably, post-stimulation of the trigeminal ganglion, researchers have observed structural changes in the dura mater, including mast cell degranulation, a process by which mast cells release granules rich in histamine and other molecules, and modifications in postcapillary venules, including platelet aggregation [86].

While it is widely accepted that such changes—particularly the initiation of a sterile inflammatory response—would be likely to cause pain, it remains uncertain whether these alterations alone are sufficient or whether they necessitate the presence of other stimulators or promoters. One of the limitations of neurogenic dural plasma extravasation as a theory is its inability to predict whether novel therapeutic targets would be effective in either the acute or preventative treatment of migraine. Indeed, the blockade of neurogenic plasma protein extravasation (PPE) has not been proven to be a reliable indicator of antimigraine efficacy in humans. This observation is substantiated by the unsuccessful outcomes of clinical trials of several potential treatments such as substance P, neurokinin 1 receptor antagonists, specific PPE blockers, CP122,288 and 4991w93, an endothelin antagonist, a neurosteroid, and an inhibitor of the inducible form of nitric oxide synthase (iNOS) named GW274150 [87].

## 5. Investigations into Neuropeptides

Through the application of electrical stimulation to the trigeminal ganglion, observable increases in extracerebral blood flow and the local release of calcitonin gene-related peptide (CGRP) and substance P (SP) have been noted in both human and cat subjects. In felines, such stimulation not only enhances the cerebral blood flow but also prompts the release of vasoactive intestinal polypeptide (VIP), a potent vasodilator peptide, via the greater superficial petrosal branch of the facial nerve. Intriguingly, the VIP ergic innervation of cerebral vessels is mainly anterior as opposed to posterior, which may make these regions more susceptible to spreading depression, possibly accounting for the common posterior onset of aura symptoms. A more specific pain-inducing area, the superior sagittal sinus, when stimulated, raises the cerebral blood flow and jugular vein CGRP levels. In human studies, elevated CGRP levels have been observed during the headache phase of severe migraine, though not in less intense attacks, as well as in cluster headaches and chronic paroxysmal hemicranias, corroborating the hypothesis that the trigeminovascular system might serve a protective function in such conditions. Migraine triggered by nitric oxide (NO) donors, which mimic typical migraine, also lead to CGRP increases that can be blocked by sumatriptan, as is the case in spontaneous migraine. Significantly, certain compounds that have been proven ineffective for migraine treatment, such as the conformationally restricted analogs of sumatriptan, CP122,288, and zolmitriptan, 4991w93, were also unable to inhibit CGRP release following superior sagittal sinus stimulation in cats. The development and successful trials of specific non-peptide CGRP receptor antagonists underscore the significance of this as a novel principle in the treatment of acute migraine. Nonetheless, considering the variability, it is unlikely to serve as a reliable migraine biomarker. Also, the lack of effect of CGRP receptor antagonists on plasma protein extravasation (PPE) explains, in part, why this model has not successfully translated into human therapeutic strategies [88].

Migraine triggers in patients include cAMP-mediated mechanisms via cilostazol, even when the CGRP receptor is blocked with erenumab. Additionally, cranial artery dilation from cilostazol remains unaffected by CGRP receptor blockage. These insights imply that migraine attacks induced by cAMP do not need CGRP receptor activation, hinting at potential novel avenues for mechanism-based migraine drug development [89].

In the realm of preclinical research, there is evidence suggesting that PACAP-specific active transport systems cross the blood–brain barrier (BBB) [90]. Yet, after crossing the BBB, PACAP isoforms either degrade quickly or re-enter the bloodstream, pointing to its primary peripheral effect. In vitro data [91] highlighted PACAP38's ability to relax vascular smooth muscle cells post-abluminal application, but not after luminal application

in cerebral arteries. In contrast, in vivo tests showed no significant change in regional cerebral blood flow due to PACAP38 intravenous infusion [93].

Throbbing headaches during migraines probably stem from pain signals from both intra- and extracranial (when vasodilated) vessels, especially arteries. No studies have focused on selective VIP blockage for migraine treatment yet. However, recent findings hint that prolonged vasodilation from VIP might induce migraine-like episodes, suggesting that VIP blockage could be a promising migraine treatment [94].

The receptors (AM1, AM2, or CGRP) that might mediate migraine-like reactions due to adrenomedullin remain unidentified [95]. AM22-52, the only known adrenomedullin antagonist, appears limited in its ability to antagonize adrenomedullin effects in rat cells, though it has shown potential in inhibiting CGRP effects [96]. No specific treatments to counteract adrenomedullin or its receptors exist currently. However, an in vitro research [97] indicated that the CGRP-receptor targeting antibody erenumab and the CGRP-receptor antagonist telcagepant opposed not only CGRP but also adrenomedullin signaling at the CGRP receptor.

Research on arresting the NO-cGMP cascade for drug development shows potential. A mouse study [56] illustrated that the sGC stimulator VL-102 induced both acute and prolonged hyperalgesia. This effect was blocked by the sGC inhibitor (ODQ) and by several antimigraine drugs (sumatriptan, topiramate, and propranolol).

Lastly, migraine patient studies have identified mutations in the TRESK potassium channel. TRESK works by inhibiting TREK1 and TREK2, amplifying the TG's excitability. As mentioned in [59,60], decreasing TG excitability using the TREK1/TREK2 agonist ML67-33 countered an NO donor-triggered migraine-like phenotype in mice similarly to the CGRP receptor antagonist olcegepant. Furthermore, it entirely reversed TG-induced facial allodynia in rats due to NO donors.

Our understanding of alternative targets leading to intracranial artery vasodilation is expanding. However, we have yet to develop successful therapies to tackle CGRP-independent mechanisms. For instance, an antibody designed to inhibit PACAP (a peptide part of the VIP, secretin, and glucagon superfamily) was developed as an alternative treatment but did not show efficacy in trials. Additionally, the results from attempts to block NO-induced reactions have been mixed. Using glibenclamide did not alleviate headaches caused by PACAP38 and levcromakalim. Still, TRPV1 agonists like capsaicin and civamide demonstrated some effectiveness due to their capacity to desensitize nerve endings hosting these channels. There is a pressing need for more research to craft alternative targeted migraine therapies, such as those focusing on VIP, amylin, adrenomedullin, PDE3, PDE5, calcium channels, and ASICs. In theory, targeting the most downstream elements like KATP channels, being the cascade's "final link", might yield better results; however, this could also bring about severe and unwanted side effects. The high induction rate of levcromakalim possibly being a result of this remains to be confirmed [57].

Changes in the connection between the hypothalamus and brainstem with the spinal trigeminal nuclei and the dorsal rostral pons have been observed during the premonitory phase of a migraine, lasting up to 48 h before pain begins [108]. The exact process that makes the hypothalamus 'overactive' in migraine situations, leading to the sensitization of trigeminal nociceptors, remains undefined. Moreover, the hypothalamus houses chemosensitive neurons that can recognize metabolic alterations in the brain and the body. External stimuli causing disruptions in balance and the brain's inherent biorhythm could potentially push the brain toward a migraine episode through hypothalamic activation [109].

NSAIDs are a reliable choice for managing acute migraine flare-ups, but care must be taken due to potential side effects like stomach issues and kidney problems. Beta-blockers serve as effective preventative measures against migraines, but they come with their own drawbacks, such as causing dizziness and fatigue. Moreover, they are not recommended for patients with specific health conditions like asthma, heart failure, and certain cardiac rhythm disorders. While calcium channel blockers have been considered for migraine prevention, the current evidence does not strongly support their use for this purpose. Anti-

seizure medications, like topiramate and divalproex sodium, and certain antidepressants, namely, venlafaxine and amitriptyline, have been found effective in preventing migraine attacks, but users must be wary of associated side effects. In deciding on a treatment course, it is crucial to weigh the potential benefits against the risks. Open dialogue between the patient and the physician will ensure the most suitable therapeutic choice is made [110].

## 6. Headache Physiology: Central Connections and the Trigeminocervical Complex

### 6.1. Migraine Neuronal Activation and Therapeutic Implications

Utilizing Fos immunohistochemistry, researchers have been able to discern the activation of cells by detecting Fos protein expression within the trigeminocervical complex. Following the irritation of meningeal with blood, there was a significant upregulation of Fos expression within the trigeminal nucleus caudalis. Additionally, upon stimulation of the superior sagittal sinus, Fos-like immunoreactivity was observed not only in the trigeminal nucleus caudalis but also in the dorsal horn at the C1 and C2 levels in both feline and simian subjects [58]. These findings are consistent with results obtained from 2-deoxyglucose analyses in congruent experiments [65]. Similarly, the activation of the greater occipital nerve, an offshoot of the C2, amplifies the metabolic activity in the aforementioned regions. It has been documented in animal-based studies that it is feasible to directly obtain readings from trigeminal neurons receiving input from both the supratentorial trigeminal and the greater occipital nerve. A mere 5 min stimulation of the greater occipital nerve resulted in a pronounced escalation in response to supratentorial dural stimuli, with the effects lasting for more than 60 minutes [101]. Conversely, the stimulation of the dura mater of the middle meningeal artery utilizing mustard oil as a C fiber irritant augmented the responses to occipital muscle stimulation [102]. Additional data derived from the Fos technique posit that such interactions likely necessitate the activation of the NMDA subtype of glutamate receptors [103]. Taken together, these findings suggest that the cervical and ophthalmic inputs intersect at the level of the second-order neuron [104]. It is worth nothing that bilateral Fos expression was observed when a lateralized structure, specifically the middle meningeal artery, was stimulated in both feline and simian models [105]. This particular group of neurons from the superficial laminae of the trigeminal nucleus caudalis and C1/2 dorsal horns is functionally recognized as the "trigeminocervical" complex. Such insights indicate that the transmission of nociceptive information from the trigeminovascular system predominantly occurs via the most caudal cells, which provides an anatomical elucidation for the referral of migraine-associated pain to the posterior cranial region. It is imperative to highlight that pharmacological experimentation has unveiled that migraine-abating drugs, such as ergot derivatives, acetylsalicylic acid, sumatriptan, eletriptan, naratriptan, rizatriptan, zolmitriptan, and novel CGRP receptor antagonists, have the potential to modulate these second-order neurons, thereby decreasing their activity [106]. This proposes another plausible avenue for therapeutic interventions in migraine. The modus operandi of triptans is believed to engage the 5-HT1B, 5-HT1D, and 5-HT1F receptor subtypes, which correlates with the positioning of these receptors on peptidergic nociceptors.

Subsequent exploration into neuropeptides has underscored the potential of CGRP receptor antagonists for migraine treatment. Elevated levels of CGRP and SP subsequent to trigeminal ganglion stimulation, observed in both humans and felines, might serve as a protective mechanism in severe migraine, cluster headaches, and chronic paroxysmal hemicranias. Additionally, an upswing in CGRP levels was documented during a NO donor-triggered migraine episode, but this surge was mitigated by sumatriptan, further corroborating the implication of the trigeminovascular system in these conditions. Nevertheless, specific compounds, such as CP122,288 and 4991w93, which were ineffective against migraine, failed to inhibit CGRP release post-superior sagittal sinus stimulation in feline subjects. In spite of these observations, the advent and ensuing triumphant clinical trial outcomes of particular CGRP receptor antagonists for acute migraine have emphasized the significance of this therapeutic strategy [64,107].

In studies involving the trigeminocervical complex, Fos immunohistochemistry has been utilized to identify activated cells by mapping the Fos protein expression. This method revealed an increased Fos expression in the trigeminal nucleus caudalis after meningeal irritation with blood and in the trigeminal nucleus caudalis and the C1 and C2 levels in the dorsal horn of cats and monkeys after stimulation of the superior sagittal sinus [111]. Moreover, the stimulation of the greater occipital nerve, a branch of C2, resulted in increased metabolic activity in these regions [112]. This suggests that inputs from the cervical and ophthalmic regions converge at the level of the second-order neuron. Pharmacological research indicates that drugs like ergot derivatives, acetylsalicylic acid, sumatriptan, eletriptan, naratriptan, rizatriptan, zolmitriptan, and CGRP receptor antagonists may help reduce cell activity at these second-order neurons and therefore could be a potential therapeutic approach for migraine [88].

Studies into serotonin–5 HT1F receptor agonists and their relation to migraine have shown that some triptans, including naratriptan, are also potent 5 HT1F receptor agonists. This suggests that 5 HT1F activation could potentially inhibit trigeminal nucleus Fos activation and neuronal firing in response to dural stimulation without affecting cranial vascular effects. These findings further support the idea that vascular mechanisms are not necessarily required for acute migraine treatments [113].

Glutamatergic transmission in the trigeminocervical complex has also been explored as a potential target for antimigraine drugs. The family of glutamate receptors (GluRs) is particularly interesting, with studies showing that NMDA receptor channel blockers can reduce nociceptive trigeminovascular transmission in vivo. Furthermore, the AMPA/kainate receptor antagonists CNQX and 2,3-Dioxo-6-nitro-1,2,3,4-tetrahydrobenzoquinoxaline-7-sulfonamide decreased Fos protein expression after the activation of structures involved in nociceptive pathways [114]. Notably, the iGluR5 kainate receptors may play a role in trigeminovascular physiology, suggesting a potential target for future treatments. In clinical trials, the iGluR5 kainate receptor antagonist LY466195 demonstrated efficacy in acute migraine treatment, reinforcing the pursuit of glutamate targets for treatment, albeit with caution regarding potential side effects [115].

*6.2. Discussion of Key Pathological Processes Involved in Migraine*

6.2.1. Neurophysiology of Migraine as a Backdrop to Imaging

The utilization of neurophysiological techniques in migraine patients has yielded significant knowledge about the condition. These techniques prioritize time resolution over spatial resolution and, until the advent of MRI, and to some extent even now, provided a higher chance for repeated trials. Research across the visual, somatosensory, auditory, and nociceptive domains has consistently shown activation patterns that differ markedly from non-migraine. These findings have led to the theory that thalamocortical dysrhythmia plays a significant role in migraine pathophysiology [116,117] (see Figure 1).

An intriguing observation from these studies is the abnormal habituation in migraine patients between attacks, as exemplified by the increased intensity of auditory evoked potentials in these periods [118]. Interestingly, this abnormality seems to normalize just before a migraine attack [119]. It is worth noting that this metric appears to be serotonin-dependent and can be modulated by triptans, which are serotonin 5-HT1B/1D receptor agonists [120]. The amplification of the passive "oddball" auditory event-related potential and an interictal habituation deficit measured by the nociceptive blink reflex, further suggest that the brain of a migraineur does not habituate in the same way as a non-migraineur's brain does [121].

These observations have given rise to the idea that the brain of a person with migraine reacts more intensely, rather than simply being hyperexcitable [121].

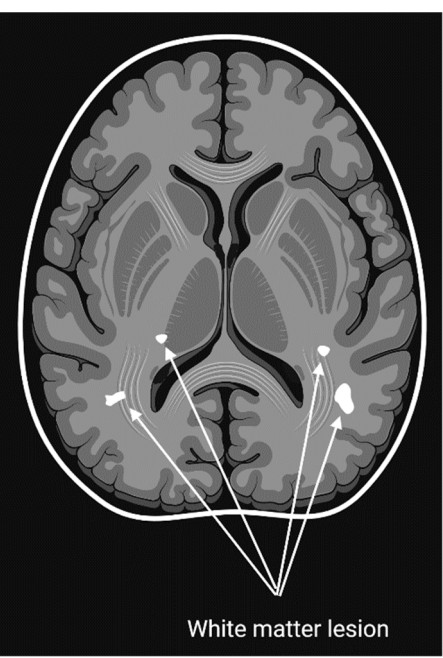

White matter lesion

**Figure 1.** The two main types of lesions found in migraine include: white matter hyperintensities and silent brain infarcts.

### 6.2.2. Inter-Attack Imaging Studies

1.   Structural Studies

Numerous research endeavors have revealed structural disparities between the brains of those who suffer from migraine and those who do not. Structural investigations, often cross-sectional, should be analyzed with consideration to the functional interactions of pain-processing areas and the trigeminal system. Voxel-based morphometry has showcased decreased grey matter in areas involved in pain processing, such as the anterior cingulate cortex, amygdala, insula, operculum, and the frontal, temporal, and precentral gyri. Interestingly, grey matter reduction in the anterior cingulate cortex has been found to be correlated with the frequency of migraine episodes [122]. On the other hand, a greater grey matter volume was seen in the bilateral caudate nuclei in high-frequency migraine sufferers compared to those with low-frequency migraine [123]. Additionally, there was an observed thickening in the somatosensory cortex, particularly in the portion responsible for mapping sensations from the head and face. This thickening was more pronounced compared to individuals who did not exhibit these changes and served as controls [124]. Research involving diffusion tensor imaging has revealed decreased fractional anisotropy in the thalamocortical pathway of individuals with migraine. Specifically, this reduction was observed in the ventral trigeminothalamic tract for those experiencing aura, and in the ventrolateral periaqueductal gray (PAG) for those without aura [124]. Other studies found only minor diffusivity changes in grey matter, while changes in white matter and brain volume were similar in both groups. Nevertheless, a more extensive investigation discovered that individuals with migraine accompanied by aura exhibited a shorter T1 relaxation time in the thalamus compared to those without aura and individuals without migraine who were in good health. In a separate study, a reduction in brain volume was detected across multiple regions when comparing migraine sufferers to control subjects. Significantly, it remains uncertain whether these alterations stem from recurrent migraine episodes or are intricately connected to the underlying mechanisms of migraine. It is worth noting that certain changes tend to revert to a more typical state during a migraine, implying a potential connection between the most recent attack and these structural modifications [125]. Taken together, these findings suggest that the structural alterations seen in migraine patients, particularly in the anterior cingulate cortex and the trigeminal somatosensory system,

reflect the brain's ability to develop migraine attacks and may underlie the progression of the disorder.

2.     Functional Studies

Functional studies serve to supplement structural brain imaging, focusing on the resting migrainous brain and its response to external stimuli.

(A) Metabolism and Receptor Pharmacology

Functional differences between two groups at rest can be evaluated using 18F-FDG PET to assess regional brain metabolism. Kim et al. [126] found that migraine was associated with reduced metabolism in central pain processing areas, suggesting a dysfunction of central pain processing in the interictal state. No area showed hypermetabolism.

(B) Stimulated Blood Flow Changes

Photophobia, a common non-head pain symptom in migraine, can result in light being perceived as overly bright or painful. Individuals suffering from migraine, even when not experiencing an attack, have been shown to tolerate less luminance than healthy individuals [127]. A study using H215O-PET revealed that exposure to different light intensities activated the visual cortex in migraine but not in controls. Moreover, applying trigeminal pain activated the same areas in control subjects, suggesting a facilitation of the retino-geniculate-cortical pathway of visual processing and/or a dysfunction of visual association areas causing photophobia. Other studies found differences in response to heat stimuli and painful heat stimulation in migraine compared to controls. Demarquay et al. [128] evaluated olfactory processing in migraine and found unique cortical responses associated with olfactory hypersensitivity. These findings suggest that the brains of migraine respond differently to external stimuli compared to healthy controls, possibly due to pre-existing functional abnormalities that worsen during a migraine attack, leading to a "dys-excitable" state [129].

(C) Studies focused on resting-state brain activity

Resting-state studies offer a unique perspective on brain function, especially in the context of disorders like migraine. These investigations focus on how the brain operates when it is not performing any particular task, thereby providing insights into the intrinsic communication patterns within the brain. They represent a significant departure from other neuroimaging techniques as follows: while structural brain imaging helps identify disparities in grey and white matter, and stimulus-driven functional magnetic resonance imaging (fMRI) pinpoints distinct dysfunctional areas, resting-state studies evaluate the interaction, or "cross-talk", among different brain regions [121].

One such study by Mainero et al. found that individuals suffering from migraine exhibit heightened connectivity between the periaqueductal gray (PAG) and multiple areas significant to nociceptive and somatosensory processing. These findings were further linked with the frequency of migraine episodes, signifying the pathophysiological relevance of this enhanced connectivity in the modulation of pain during migraine [130].

Another vital aspect of migraine is the presence of cutaneous allodynia, considered a reflection of central sensitization during migraine attacks. When comparing resting-state connectivity between migraine with and without cutaneous allodynia, distinctive patterns were identified. Specifically, connectivity differences were noted between the PAG/nucleus cuneiformis and various discriminative pain processing centers, such as the brainstem, thalamus, insula, cerebellum, and higher-order pain-modulating areas located in frontal and temporal regions [131]. This evidence suggests that individual symptoms during a migraine attack might be determined by abnormal communication between pain-modulating areas during interictal periods.

While all participants in the study had migraine with normal routine investigation results, the presence of ictal allodynia seems to delineate different subtypes of migraine. This implies that migraine itself could also be pathophysiologically diverse. In terms of headache phase studies, seed-based resting-state fMRI has exhibited increased connectivity between primary visual and auditory cortices and the right dorsal anterior insula, and between the dorsal pons and the bilateral anterior insulae. Interestingly, these findings

did not correlate with migraine frequency, suggesting that these changes were inherent characteristics of migraine pathophysiology rather than episodic manifestations [121].

Expanding the scope, resting functional connectivity of certain brain regions, including the right middle temporal, posterior insula, middle cingulate, left ventromedial prefrontal, and bilateral amygdala regions, was found to effectively distinguish between the brains of migraine and non-migraine [132]. Despite being grounded in clinical observation, these techniques have the potential to yield insights into migraine biology, contribute to the development of NextGen treatments, and offer biomarkers of change, particularly for preventive studies.

Resting-state studies are not limited to the seed-based approach, which focuses on the connectivity of specific "seed" areas, such as the PAG or nucleus cuneiformis. Task-free resting-state studies employing independent component analysis without a priori hypothesis have also been conducted. One such study by Tessidore et al. [133] examined the default mode network in patients with migraine without aura. They found decreased connectivity in the prefrontal and temporal regions of the default mode network among these patients. The authors speculated that this could indicate a dysfunction of the default mode network, potentially tied to maladaptive responses to stress or environmental triggers, which are often characteristic of migraine.

In conclusion, being a migraineur suggests the presence of nuanced differences in brain structure and function, even outside of active migraine attacks. Notably, most areas showing such differences belong to the non-specific pain processing areas or the trigeminal system. A significant challenge moving forward is understanding how these differences predispose individuals to migraine, and identifying which structures drive the transition from the interictal phase, through the premonitory phase, to the headache phase, and eventually, the postdrome period that returns to the interictal phase.

(D) Studies focusing on mitochondrial energy metabolism

Considering the recognition of migraine as a component of mitochondrial cytopathies, exploring how this biological aspect influences the onset and development of migraine provides a promising research avenue. Although initial studies on mitochondrial DNA did not find any typical MELAS or MERRF mutations, a successfully conducted randomized controlled trial of riboflavin (vitamin B2) as a preventive treatment for migraine supports the hypothesis that metabolic dysfunction could increase susceptibility in some patients [134].

Utilizing 31P-NMR spectroscopy, Welch et al. [135] identified changes in the phosphate metabolism in patients with migraine with aura during an attack. Later studies using the same technique found similar metabolic shifts in patients with migraine without aura and even in children [136]. Furthermore, by employing 3T MRI and 31P-NMR spectroscopy, researchers were able to identify alterations in energy metabolism in the occipital cortex of patients with migraine without aura [137]. Given the observed variations in energy changes among patients, these findings could potentially explain some, but not all, of the biological mechanisms contributing to the manifestation of migraine.

## 3. Premonitory Phase Studies

From a clinical perspective, the premonitory phase—the transitional period between the asymptomatic interictal phase and the onset of a headache attack—is crucial for understanding what triggers migraine. An essential fMRI study in this regard examined the activation and deactivation patterns induced by the trigemino-nociceptive stimulation of the nasal mucosa as the day of the headache approached [138]. In comparison to control subjects, interictal migraine showed reduced activation of the spinal trigeminal nuclei. Interestingly, this deactivation demonstrated the following cyclic behavior throughout a migraine interval: normalization prior to the next attack and a significant reduction of deactivation during the attack. This cyclical behavior may reflect the brain's increased susceptibility to initiate the next attack, with the identification of its pacemaker critical to our understanding of the initiation of a migraine attack.

Clinically, the earliest indicators of an impending migraine attack are known as premonitory symptoms, which manifest before the onset of head pain and signal to the patient

that a headache is imminent. These symptoms, likely tied to the hypothalamus [139], include concentration problems, fatigue, irritability, and depression. A recent study by Maniyar et al. [140] induced migraine attacks in eight patients who could predict the onset of a headache by a pronounced premonitory phase. During this phase, which occurs before the onset of head pain, H215O-PET showed activation of the hypothalamus, midbrain ventral tegmental area, and the PAG. This functional representation of premonitory symptoms hints at the potential role of the hypothalamus in triggering migraine. Additional data from a single patient tracked with BOLD-fMRI over a 30-day period showed increased hypothalamic responses as the attack approached, and the effects were coupled with the dorsolateral pons [141]. Additionally, the hypothalamus could play a crucial role in non-headache symptoms during the pain phase since its activation was observed in spontaneous migraine attacks using H215O-PET [142]. Interestingly, activations reported in trigeminal-autonomic cephalalgias are more posterior than those reported in migraine [121].

## 4. The Aura Phase

Typically, a visual aura in the context of a migraine presents itself before the onset of the headache phase, although there are instances where it coincides with the headache or even occurs without any headache at all. The manifestation of this aura often begins as a scintillating or blind spot situated in the center of the individual's field of vision [121]. Personal experiences reported by Lashley [143] indicated that this visual disturbance or scotoma progressively expanded over a period of approximately one hour, moving in a C-shaped trajectory towards one side's temporal visual field. Based on his observations, the estimated speed of this phenomenon over the visual cortex was calculated to be approximately 3 mm per minute.

A few years later, the concept of a potential underlying mechanism emerged from the work of Leão, who stimulated the cortices of rabbits electrically. He observed a depression in the electroencephalogram (EEG) readings that spread out from the stimulation site at a similar speed of 3 mm per minute. Leão postulated that this cortical phenomenon could possibly serve as the foundation for the migraine aura [144]. This theory sustained for several decades that the occurrence of a typical visual aura could be associated with this phenomenon, termed "cortical spreading depression" (CSD) [145].

The validation of the CSD occurrence in humans was conjectural until a groundbreaking study by Olesen et al. [146]. They injected Xenon-133 into the carotid artery during a human migraine aura and found an observable progressive alteration of regional cerebral blood flow (rCBF). Fast forward two decades later, patients who could self-trigger their visual aura or who were capable of reaching a medical facility during the early stages of a visual aura were included in a functional magnetic resonance imaging (fMRI) study using checkerboard stimulation. The change in blood oxygenation level-dependent (BOLD) signal in the visual cortex in response to checkerboard stimulation during the progression of a visual aura showed characteristics similar to those of the CSD observed in animal models. This included a signal spread at a velocity of roughly 3.5 mm per minute, aligning with the earlier clinical predictions and the CSD observed in rabbit cortices [147]. These findings suggest that the visual aura experienced in migraine might indeed be the result of a CSD-like event. Additionally, the study by Hadjikhani et al. [148] pinpointed the origin of this unique response to checkerboard stimulation to be located in the visual association cortex V3A.

## 5. The Headache Stage

Renowned as the most prominent symptom of a migraine attack, the headache stage is often the defining phase for many patients. To diagnose a migraine, however, additional symptoms need to be present. These can range from nausea, photophobia (light sensitivity), phonophobia (sound sensitivity), to sensitivity to movement [149]. The imaging patterns observed during a migraine are typically a composite of these symptoms, with some possibly reflecting individual symptoms like head pain, photophobia, or allodynia (an

increased response to pain), and others hinting at underlying mechanisms that trigger the migraine.

(a) The Experience of Head Pain: The complexity of primary headache disorders extends beyond head pain that is typically triggered by harmful stimuli on the skin. However, the sensation of pain is a shared experience in harmful head pain and spontaneous migraine attacks. Therefore, the markers identified in functional brain imaging of experimental head pain should also be observable in migraine headaches. Thus, any additional regions highlighted in primary headache disorders may provide specific insights into migraine, potentially revealing symptoms beyond head pain or even mechanisms that drive migraine attacks. Functional brain imaging of harmful pain in the head is a significant focus as it could improve our understanding of functional brain imaging of migraine. May et al. [150] used H215O-PET to measure rCBF in seven healthy subjects after injecting a small amount of capsaicin into the forehead. They noticed an increase in rCBF in several brain areas during the pain state, including the bilateral insula, the anterior cingulate cortex, the cavernous sinus, and the cerebellum. Notably, there was no activation of the brainstem.

(b) Migraine Attacks: Over the past two decades, a seminal study employing positron emission tomography sought to evaluate regional cerebral blood flow utilizing 15C-labeled O2 inhalation in a cohort of nine individuals experiencing spontaneous right-lateralized migraine episodes. Relative to the non-painful interlude, the migraine episodes were concomitant with augmented rCBF in regions, such as the cingulate cortex, auditory association cortex, and the parieto-occipital juncture proximate to the visual association cortex. Furthermore, the migraine-afflicted state manifested with escalated rCBF in the midbrain, the dorsal rostral pons adjacent to the periaqueductal gray, and the raphe nuclei [128]. Contrasting the generalized pain signature derived from the capsaicin experiment, this investigation attributed diverse migraine-related symptoms to distinct cerebral domains as follows: the experience of cephalic pain was associated with the cingulate cortex, photophobia was linked to the visual association cortex, and phonophobia was ascribed to the auditory association cortex. The cessation of these symptoms was synchronous with the waning of the aforementioned signals. Yet, the elevated rCBF in the brainstem endured during the nascent non-painful phase, implying that this anatomical region might not merely be symptomatic but could also typify a dysfunction pivotal for initiating or perpetuating a migraine episode. Clinical research, coupled with fundamental studies, further bolsters the centrality of the brainstem in the pathogenesis of migraine. An illustrative point being the emergence of migraine episodes in individuals previously devoid of migraine who underwent deep brain stimulation targeting the PAG for unrelated pain conditions. Moreover, the progressive accumulation of iron in the PAG over the disease's tenure intimates the indispensable role of the PAG in migraine genesis. Corroborating this notion, a plethora of animal-centric studies have delineated how brainstem nuclei, specifically the PAG and raphe nuclei, profoundly modulate trigeminovascular pathways in laboratory-induced migraine paradigms [91,129].

Functional brain imaging employing advanced techniques of enhanced spatial and temporal resolutions have corroborated the pivotal role of the brain stem in the pathophysiology of migraine. In a comparative study, Bahra et al. [130] distinguished migraine from cluster headaches, underscoring the specificity of brain stem activation to migraine. Examining the lateralization of this activation during unilateral migraine episodes, Afridi et al. [131] ascertained that the activation was ipsilateral to the side of the headache. This suggests the possibility that unilateral migraine might be attributed to unilateral dysfunction of the brain stem. In a preceding discourse, Maniyar et al. [117] detected activation in the dorsal rostral pons, the PAG, and the hypothalamus during the preliminary premonitory phase of migraine, buttressing the hypothesis of a central "migraine mediator" located in these regions. Melding the clinical manifestations of migraine—typified by altered sensory, nociceptive, photic, acoustic, and olfactory perceptions—with functional imaging insights, it becomes evident that either the brain stem, hypothalamic structures, or a combination of both are instrumental in migraine pathophysiology. Such structures

potentially pinpoint the anatomical epicenters of cerebral dysfunction engendering the multifaceted dynamics of migraine episodes.

(c) Photophobia: Denuelle et al. [119] conducted research on eight individuals suffering from migraine, evaluating them during headache episodes, post-sumatriptan alleviation, and interictal periods. Utilizing continuous light stimulation, they discerned that low luminance provocation elevated rCBF as indicated by H215O-PET scans. During the headache phase, hyperperfusion was detected in the cuneus, and post-relief, both the cuneus and lingual gyrus demonstrated this phenomenon; conversely, such changes were absent interictally. This might insinuate an augmented excitability of the visual cortex amidst migraine occurrences. Notably, even post-headache alleviation, the persistence of this hyperperfusion, unrelated to the headache's presence, hints at the structural foundation of photophobia potentially residing in primary and ancillary visual cortices. Moreover, zones responsive to minimal luminance during migraine were equivalently reactive interictally to escalated luminous intensities, further substantiating the cyclic nature underpinning migraine and their concomitant symptoms.

In studies targeting the premonitory phase, focusing on non-painful symptoms, pivotal insights have emerged, emphasizing the separation of such phenomena from pain while highlighting their integral role in migraine biology. When contrasting individuals with provoked premonitory symptoms, those exhibiting photophobia (or perhaps more aptly termed photic hypersensitivity due to the absence of pain) demonstrated activation within the extrastriate visual cortex, specifically Brodman area 18 [117]. Interictal connectivity within the visual system, manifested in the lingual gyrus, was also identified using resting-state methodologies. Furthermore, in experiments distinguishing migraine with and without nausea, those experiencing nausea showcased activation in the rostral dorsal medulla encompassing regions like the nucleus tractus solitarius, the dorsal motor nucleus of the vagus nerve, and the nucleus ambiguous; moreover, activation in the PAG was also evident. These research endeavors have enriched our comprehension of cerebral regions implicated in migraine, unequivocally indicating mechanisms transcending mere pain dependency [91].

Different visual migrainous phenomena are associated with dysfunctions in different areas of the visual association cortex. For instance, the cuneus and lingual gyrus are involved in photophobia, while V3A might be the origin of a typical visual aura. When comparing the imaging results during the migraine premonitory phase, such as hypothalamic and brain stem activation, with those of cortical activation during a typical migraine aura, it appears likely that the aura and migraine are distinct phenomena [121].

6.      Blood–Brain Barrier (BBB): The integrity of the BBB in migraine

In postdrome, patients often describe fatigue, difficulty with concentration, and a need for sleep. Some patients also report a feeling of elation and well-being and a return of appetite. The symptoms are not always perceived as bothersome and are commonly overlooked by patients and doctors. Often, the patient is just relieved that the headache phase is over.

More work is needed to understand the nature and cause of the postdromal phase. It would be especially helpful to understand the brain's role in postdromal phase symptoms, whether the brain goes back to normal after a migraine attack, and if not, why not. Also, more understanding of how the brain recovers and how quickly it recovers would be very helpful. Whether this phase represents a therapeutic opportunity is unknown but should be explored [121].

### 6.3. Calcitonin Gene-Related Peptide (CGRP) in Migraine

6.3.1. Role of CGRP in Migraine Development and Progression

1.      Introduction to CGRP and its significance in migraine

Calcitonin gene-related peptide (CGRP) is an incredibly potent neuropeptide comprised of 37 amino acids. It serves as a vasodilator and is produced within neurons located

in both the peripheral and central nervous systems. This neuro-peptide binds to a complex heterodimer receptor, which is primarily composed of a class B G-protein coupled receptor, commonly referred to as CLR (calcitonin receptor-like receptor) [151].

Within the central nervous system, empirical research has identified elevated levels of CGRP in the blood and saliva of patients who experience certain headache disorders. Such disorders include migraine and cluster headaches, as well as neuralgias like trigeminal neuralgia, chronic paroxysmal hemicranias, and even rhinosinusitis. It is noteworthy that the levels of CGRP remain heightened during a migraine episode and continue to be elevated in-between these attacks for patients suffering from chronic migraine. Additionally, studies have revealed that exogenous infusions of CGRP can initiate a migraine episode [152].

CGRP's role extends to the pathogenesis of migraine, which is an intricate neurovascular disorder. This is typically characterized by a throbbing or pounding headache, which affects one side of the head. It is often accompanied by other symptoms, such as photophobia (light sensitivity), phonophobia (sound sensitivity), nausea, vomiting, and even disability. Additionally, the duration of a typical migraine episode can last between 4 and 72 h [153]. Researchers have found that CGRP triggers the release of vasoactive neuropeptides in trigeminal neurons, leading to vasodilation of the cerebral vasculature, thereby contributing to the emergence of a migraine.

The Food and Drug Administration (FDA) has sanctioned several medications specifically targeting CGRP or its receptor for the management and prevention of migraine. Prominent among these are monoclonal antibodies including erenumab, eptinezumab, galcanezumab, and fremanezumab, which zero in on the CGRP receptor. In addition, CGRP receptor antagonists, like rimegepant and ubrogepant, have been incorporated into therapeutic regimens. Two other receptor antagonists, atogepant and vazegepant, remain under clinical evaluation and anticipate FDA endorsement. Prophylactic interventions for both episodic and chronic migraine commonly incorporate erenumab, eptinezumab, galcanezumab, and fremanezumab. For addressing acute migraine manifestations, with or without the presence of an aura, rimegepant and ubrogepant are the preferred choices. Presently, the efficacy of galcanezumab and fremanezumab in precluding cluster headaches is a subject of active research [135].

Interestingly, CGRP has been shown to have cardio-protective properties in pathological conditions. For instance, research conducted on rodent models of various cardiovascular diseases has revealed this beneficial action of CGRP. Human studies also support this notion by showing that CGRP can reduce afterload and increase inotropy, which are potentially cardioprotective effects, particularly in cases of heart failure. Despite these findings, no drugs have yet been developed to harness this cardio-protective effect of CGRP on the cardiovascular system [154].

Recent research has also begun to uncover CGRP's involvement in numerous other physiological and pathological phenomena. These include peripheral nerve regeneration, Alzheimer's disease, regulation of vascular tone in mesenteric arteries, and even pregnancy. Despite these promising findings, no medications have been developed to date to leverage these potential beneficial effects of CGRP [155].

Preclinical studies have demonstrated that calcitonin gene-related peptide (CGRP) exhibits activity in both the central and peripheral nervous systems (CNS and PNS), making it a crucial element in the pathophysiology of migraine. In the periphery, CGRP acts on a number of targets, such as mast cells, blood vessels, glial cells, and trigeminal afferents located in the meninges, along with neural cell bodies and satellite glia found in the trigeminal ganglia. Within the meninges, CGRP is thought to contribute to neurogenic inflammation by stimulating mast cells to release neuron-sensitizing agents. This cascade effect can lead to enhanced vasodilation in the dura. Consequently, the modulation of neural activity within the meninges may instigate a feedback loop, ultimately leading to peripheral sensitization of nociceptors [156]. The notion of CGRP playing a peripheral role in migraine is strongly supported by the effectiveness of systemically administered CGRP-targeting monoclonal antibodies, which exhibit poor permeability to the blood–brain

barrier (BBB) [157]. It is clear that peripheral sensitization is critical for CGRP's actions and likely establishes the foundation for CGRP actions in the CNS.

Within the central nervous system (CNS), the distribution of CGRP and its receptor spans various pathways postulated to play pivotal roles in migraine pathophysiology. Situated externally to the blood–brain barrier (BBB), the trigeminal ganglion extends its projections to the trigeminal nucleus caudalis (TNC). From here, second-order neurons transmit signals to the posterior thalamic area (PTA), an umbrella term denoting all nuclei within this specific thalamic region. Serving ostensibly as a hub for sensory integration, the PTA exhibits functional aberrations during migraine occurrences. Neurons in the thalamus receive inputs from both the TNC and retinal ganglion cells. Crucial rodent studies have accentuated the import of the PTA in photophobia's onset, proposing its role as a nexus for light and pain integration [139].

Situated in select nuclei of the PTA, CGRP and its receptors are postulated to be integral to this pathway—a hypothesis fortified by research showing that CGRP infusion into the PTA augments neuronal activity. Additionally, ascending pathways carrying somatosensory and nociceptive stimuli converge upon the CGRP-expressing neurons located in the subparafascicular and intralaminar nuclei. In human subjects, during migraine attacks, activation is discerned in the posterior thalamus, which also manifests altered connectivity patterns with various brain areas. Taken together, this suggests that the neuromodulatory actions of CGRP, observed in distinct neural networks, might be instrumental in rendering the PTA hyperresponsive to sensory input [140].

Further elucidating this sensory hyperreactivity, pathways potentially involve the parabrachial nucleus (PBN), colloquially termed the "general alarm" system. Functioning as an intermediary for pain and assorted sensory signals en route to the forebrain, the PBN receives direct extensions from the trigeminal nucleus, influencing the emotional facet of pain. Given the abundance of CGRP in the PBN and its extensive projections to brain regions implicated in migraine pathophysiology, modified signaling within this conduit might underlie the heightened sensory perception characteristic of migraine. CGRP's dual—peripheral and central—actions likely synergize to precipitate a migraine. Considering the disorder's multifaceted nature, it is improbable that a singular action of CGRP singularly instigates migraine [141].

2. Mechanisms by which CGRP contributes to migraine symptoms

The role of CGRP in migraine symptomatology is primarily deciphered through preclinical investigations. Recognized as a paramount instigator of migraine, CGRP can elicit an array of migraine-reminiscent symptoms in animals that parallel the effects observed when humans are infused with CGRP, encompassing pain-related symptoms. In rodents, mechanical hypersensitivity, an often-reported migraine symptom, can be instigated following CGRP administration. For instance, CGRP's dural administration in mice provoked periorbital touch hypersensitivity, whereas its intrathecal introduction led to heightened pain sensitivity in rat hindpaws and amplified mechanical allodynia in mice upon pinch [140].

For an extended period, the challenge of gauging spontaneous pain in animals persisted due to the absence of an apt assessment method. However, in 2010, Mogil and his team pioneered a method, illustrating that specific pain forms could be evaluated via facial grimace scales without necessitating an evoked response [142]. Building on this foundation, our research demonstrated that injecting CGRP peripherally in mice culminates in spontaneous, migraine-analogous pain, which sumatriptan could substantially mitigate. We employed an uninterrupted, objective appraisal of eye closure to assess the grimace induced by CGRP. Our findings also elucidated that CGRP-induced pain was not influenced by light levels, positing that pain and light aversion, another symptom induced by CGRP, operate independently [143].

Photophobia, characterized by an augmented sensitivity or discomfort in light conditions usually deemed non-painful, stands as a diagnostic hallmark of migraine. Those afflicted with migraine often perceive even subdued light as unsettling, gravitating away

from luminous environments [144]. This human experience has been adeptly mirrored in a mouse paradigm where exposure to light becomes aversive post-CGRP administration, both centrally and peripherally, in conventional mice. This aversive reaction to light can be palliated by triptans, insinuating that the murine aversion to light resonates with human migraine. Intriguingly, in a specialized mouse model sensitized to CGRP—where human RAMP1 (an essential component of the CGRP receptor) is overexpressed within the nervous system—a mere 55 lux light intensity suffices to trigger light aversion post central CGRP administration. Notably, this aversive reaction to light is not an offshoot of anxiety, as evidenced by the unaffected performance of these mice in a light-agnostic anxiety assessment (the open field test) [145,146]. Furthermore, post-CGRP injection, these mice exhibit diminished mobility, but this inertia is predominantly observed in dimly lit sections of their enclosure. Such a preference for darker locales and a proclivity to rest mirrors human behavioral tendencies during migraine episodes.

### 6.3.2. CGRP as a Therapeutic Target

1. Overview of CGRP-targeted treatments in migraine management—Evaluation of the effectiveness of CGRP inhibitors

Over recent years, a plethora of molecules aiming to obstruct CGRP signaling pathways have been developed, with the objective of mitigating migraine symptoms. The first molecules that showed promise were CGRP receptor antagonists, known as "gepants." These substances demonstrate a high affinity for the canonical CGRP receptor, blocking the CGRP from binding and obstructing the subsequent signal transduction. Importantly, gepants do not incite direct vasoconstriction, making them potentially safer than triptans for a migraine population that statistically exhibits a higher prevalence of cardiovascular diseases [158].

Numerous clinical trials have established that both intravenous and orally administered gepants can effectively alleviate acute migraine symptoms (Table 3); however, the efficacy of gepants in preventing migraine is currently a matter of debate. Some clinical trials had to be halted due to adverse effects, while others are still in progress [159]. The development of some gepants was ceased for various reasons. For example, olcegepant demonstrated low oral bioavailability, and both telcagepant and MK-3207 were discontinued due to liver toxicity associated with frequent use [159].

In spite of these initial safety concerns, the apparent efficacy of gepants has encouraged continued efforts to devise safe molecules that block CGRP. Currently, three gepants—rimegepant, ubrogepant, and atogepant—are still under clinical development. In phase 2b clinical trials, the efficacy of rimegepant for acute migraine treatment was assessed using various endpoints, such as freedom from pain, migraine, photophobia, phonophobia, and nausea remission [76]. Medium doses of rimegepant (75, 150, and 300 mg) were found to be significantly more effective than the placebo, and unlike the previously terminated gepants, rimegepant did not exhibit any adverse effects on liver function. Interestingly, a higher dose of rimegepant (600 mg) did not yield significant benefits over the placebo, leading the researchers to hypothesize that this could be attributed to inherent variability among patients randomized to this dose group [160].

Post this investigation, a series of three phase 3 double-blind, randomized, placebo-controlled trials (NCT03235479, NCT03237845, NCT03461757) alongside a safety investigation (NCT03266588) were commenced, with outcomes yet to be unveiled [150]. In parallel, ubrogepant showcased a favorable dose-effect correlation for acute migraine treatment in a phase 2b double-blind, randomized, placebo-controlled trial [151], exhibiting negligible side effects. However, these findings are somewhat overshadowed by the heightened placebo group response and the study's restricted patient count. Two subsequent phase 3 double-blind, randomized, placebo-controlled trials (NCT02867709, NCT02828020) wrapped up in December 2017 and February 2018, respectively. The initial findings echo the results of the earlier phase 2b study. Atogepant, possessing a molecular structure distinct from its gepant counterparts, is currently under examination for migraine prevention. Initial data from a

phase 2b/3 trial (NCT02848326) indicate that adults administered atogepant underwent a more pronounced decline in their monthly migraine days average compared to those receiving a placebo. There were no reported grave side effects tied to the treatment. As of the time this review was drafted, ubrogepant emerged as the inaugural gepant to secure FDA sanctioning for acute migraine intervention, encompassing cases with or devoid of an aura [140].

**Table 3.** Clinical trials investigating drugs that target CGRP (calcitonin gene-related peptide): both completed and currently underway.

| Drug Name, Type of Molecule | Indication (Acute or Prophylactic) | Development Stage |
| --- | --- | --- |
| Atogepant, CGRP antagonist | Prophylactic | FDA-approved |
| BI 44370 TA, CGRP antagonist | Acute | Abandoned |
| Eptinezumab, CGRP monoclonal antibody | Prophylactic | FDA-approved EMA-approved |
| Erenumab, CGRP receptor monoclonal antibody | Prophylactic | FDA-approved |
| Fremanezumab, CGRP monoclonal antibody | Prophylactic | FDA-approved |
| Galcanezumab, CGRP monoclonal antibody | Prophylactic | FDA-approved |
| MK-3207, CGRP antagonist | Acute | Abandoned for liver toxicity |
| Olcegepant, CGRP antagonist | Acute | Abandoned for lack of oral availability |
| Rimegepant, CGRP antagonist | Acute | FDA-approved |
| Telcagepant, CGRP antagonist | Acute and prophylactic | Abandoned for liver toxicity |
| Ubrogepant, CGRP antagonist | Acute | FDA-approved |

Monoclonal antibodies aimed at CGRP (such as fremanezumab, galcanezumab, and eptinezumab) and its receptor (like erenumab) form a separate molecular category adept at obstructing CGRP signaling pathways. Three among these antibodies (fremanezumab, galcanezumab, and erenumab) recently achieved FDA endorsement for preventive migraine therapy, while a verdict on a fourth candidate, eptinezumab, is anticipated in 2020. Remarkably, about half of the patients administered these antibodies witnessed a 50% downturn in their migraine days. Notably, no discernible efficacy disparity was observed across antibodies, irrespective of whether they latch onto the receptor or isolate CGRP. These antibodies also maintain their therapeutic effectiveness for an extended period beyond a month post-application, thus qualifying as prophylactic agents administered monthly or even on a quarterly schedule to patients. This mode of application stands in stark contrast to the daily oral dosing demanded by gepants [140].

Drawing from extensive clinical trials and nearly a year's presence in the market, CGRP and its receptor antibodies seem to have a good safety profile and are generally well-tolerated. Yet, the ramifications of the prolonged CGRP blockade remain to be understood. A glimmer of optimism emerges from Amgen/Novartis's findings, which indicate that their antibody remained safe up to the three-year mark in an ongoing five-year open-label study [152]. Moreover, an examination focusing on patients diagnosed with angina did not highlight any detrimental effects of the antibody [153]. Still, this study is not without its constraints. It predominantly featured male participants in a disorder that chiefly affects females, roped in patients with stable angina pectoris instead of those with microvascular disease (who would better mirror the vulnerable population), and gauged

drug implications rather prematurely before the receptor antibody could adequately attach to the receptor (see Figure 2) [154].

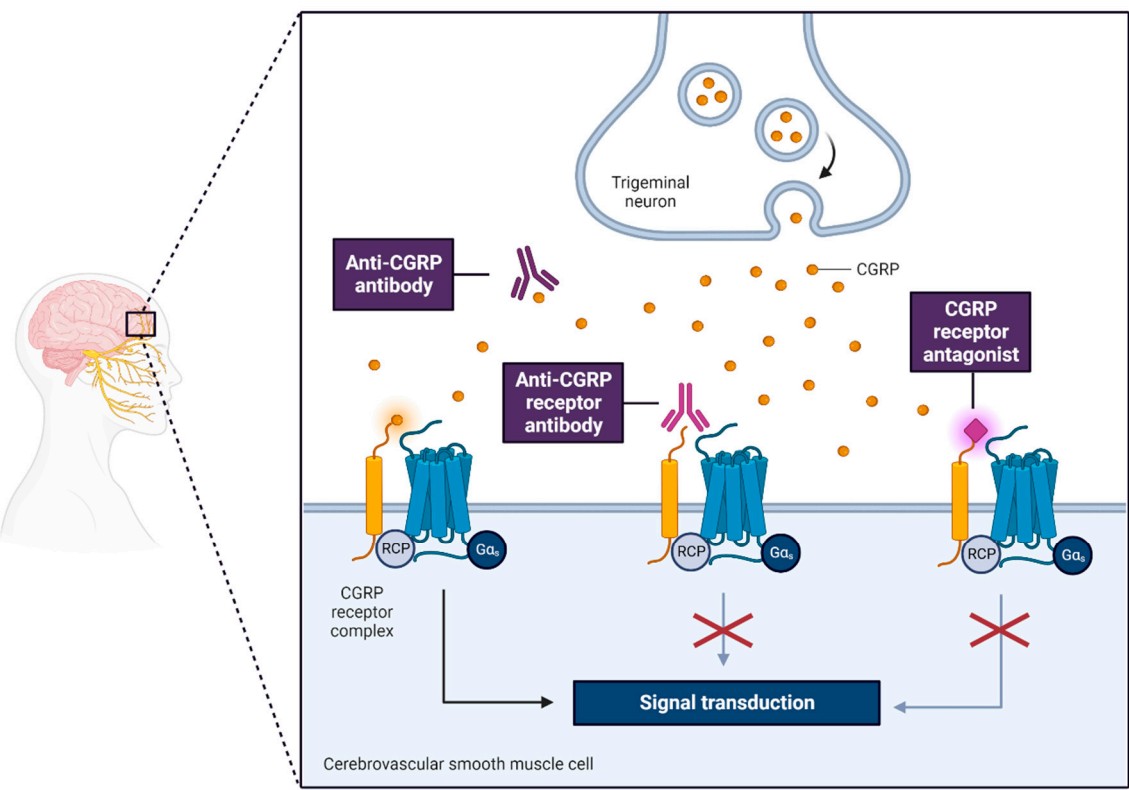

**Figure 2.** Calcitonin gene-related peptide-targeting drugs for migraine.

Cardiovascular risks loom large, especially considering that migraine patients are identified as high-risk candidates for stroke and cardiovascular complications. The following pressing question arises: might a CGRP blockade amplify the intensity of a stroke episode? A recent case study highlighted a patient who encountered an ischemic incident post the administration of a CGRP antagonist medication [155]. Still, it is pivotal to understand that conclusions can not be hastily drawn from a solitary patient's experience. Rigorous, long-duration studies centered on cardiovascular wellness are imperative. A prudent starting point would be animal-based research endeavors, delving into the impacts of CGRP blockade on ischemic conditions (see Figure 2).

### 6.4. Medical Treatment of Migraine

6.4.1. Overview of Conventional Medical Treatments for Migraine

1. Medications commonly prescribed for migraine relief—Discussion of their mechanisms of action and limitations (Figure 3)

**Therapeutic Interventions for Migraine Cessation**

(a) Anti-Inflammatory Agents: NSAIDs and Acetaminophen

Non-steroidal anti-inflammatory drugs (NSAIDs) serve as the primary selection for mitigating the intensity and duration of migraines and are supported by an extensive body of evidence. Various NSAIDs, such as ibuprofen, naproxen sodium, acetylsalicylic acid (ASA), diclofenac potassium, aspirin, tolfenamic acid, piroxicam, ketoprofen, and ketorolac, have all exhibited their efficacy in treating migraine through evidence gleaned from randomized controlled trials and systematic reviews. Acetaminophen, as well as a combination formula of acetaminophen, aspirin, and caffeine, has also displayed significant efficacy in the acute treatment of migraine [161].

Mechanism of Action

NSAIDs primarily act by hindering the synthesis of prostaglandins. They act to reversibly block cyclooxygenase (COX) enzymes 1 and 2. The NSAIDs that inhibit prostaglandin E2 synthesis are particularly effective in mitigating acute migraine attacks. Aspirin, for instance, serves as an irreversible inhibitor of both COX 1 and 2 enzymes.

The complete mode of action of acetaminophen is not fully understood yet; however, it is believed to exert its effects on central processes, like enhancing the serotonergic descending inhibitory pathways. Acetaminophen may also interact with opioidergic systems, eicosanoid systems, and the nitric oxide-containing pathways [162].

(b) Triptans

The U.S. Food and Drug Administration (FDA) has given the green light to a total of seven triptans specifically designed for the immediate relief of migraine episodes. This lineup includes sumatriptan, eletriptan, naratriptan, zolmitriptan, rizatriptan, frovatriptan, and almotriptan. In terms of pricing, triptans tend to be on the higher side compared to NSAIDs. As a result, they are generally selected as a treatment strategy when alternatives like NSAIDs or acetaminophen do not produce the desired outcomes, or when the intensity of the migraine demands their application [158].

Mechanism of Action

Triptans function as serotonin-receptor agonists. They possess high affinity for the 5-HT1B and 5-HT1D receptors, and they have variable affinity for the 5-HT1F receptors. The supposed mode of action involves binding to postsynaptic 5-HT1B receptors located on the smooth muscle cells of blood vessels and to presynaptic 5-HT1D receptors situated on trigeminal nerve terminals and dorsal horn neurons [163].

(c) Antiemetics

Antiemetics are frequently chosen for migraine treatment when symptoms include nausea or vomiting. These medications can be administered either alongside NSAIDs or triptans, or used as monotherapy. Metoclopramide and prochlorperazine are two commonly employed antiemetics. Metoclopramide has the most significant body of evidence supporting its efficacy in treating migraine and is less likely to cause extrapyramidal side effects compared to prochlorperazine. Other antiemetics used for migraine management include domperidone, promethazine, and chlorpromazine [161].

Mechanism of Action

Metoclopramide is a benzamide that antagonizes the D2 receptor at lower doses and the 5HT-3 receptor at higher doses, providing both antiemetic and migraine relief effects. Both prochlorperazine and chlorpromazine function as dopamine antagonists, interacting with the D2 receptor, which helps in mitigating the symptoms of migraine and controlling nausea and vomiting.

(d) Ergotamines

With the advent of triptans, the usage of ergotamines has declined as triptans have shown superior efficacy. Dihydroergotamine has shown some efficacy in treating migraine, while the effectiveness of ergotamine remains unclear. A systematic review revealed that dihydroergotamine was not as effective as triptans; however, when combined with an antiemetic, dihydroergotamine was found to be as effective as ketorolac, opiates, or valproate [164]. Dihydroergotamine might be a useful alternative when patients do not respond to other medications, including triptans.

Mechanism of Action

Ergotamines, similar to triptans, are potent agonists of 5-HT 1b/1d receptors. Their mechanism of action is thought to involve the constriction of the presumed pain-causing intracranial extracerebral blood vessels at the 5-HT1B receptors and inhibition of trigeminal neurotransmission at both peripheral and central 5-HT1D receptors; moreover, they interact with other serotonin, adrenergic, and dopamine receptors. They induce the constriction of peripheral and cranial blood vessels [165].

**Interventions Aimed at Migraine Prevention**

(a) Beta-Blockers

Beta-blockers, such as propranolol, timolol, bisoprolol, metoprolol, atenolol, and nadolol, have been explored for their prophylactic role in preventing migraine attacks and have shown positive results in clinical studies. However, beta-blockers exhibiting intrinsic sympathomimetic activity, like acebutolol, alprenolol, oxprenolol, and pindolol, do not appear to demonstrate efficacy for migraine prevention [166].

Mechanism of Action

The exact mechanisms underlying the preventive effect of beta-blockers on migraine are not completely understood. One prevailing theory suggests that their migraine prevention abilities may be linked to their beta-1 mediated effects, which inhibit the release of noradrenaline and activity of tyrosine hydroxylase, thereby contributing to their prophylactic action. Other potential mechanisms may involve the serotonergic blockade, inhibiting thalamic activity, and blocking the effect of nitrous oxide.

(b) Antiepileptic Drugs

Several antiepileptic drugs (AEDs) have been investigated for their efficacy in migraine prevention, with topiramate and valproate demonstrating the most substantial evidence of effectiveness [166].

Mechanism of Action

The precise mode of action of antiepileptic drugs in preventing migraine remains elusive. For topiramate, it is known to block several channels, such as voltage-dependent sodium and calcium channels. In addition, it has been shown to reduce glutamate-mediated excitatory neurotransmission, enhance the inhibition mediated by GABA-A, inhibit carbonic anhydrase activity, and decrease CGRP secretion from trigeminal neurons, all of which could potentially contribute to its preventive effects on migraine. Similarly, for valproate, a multifaceted approach is likely involved in preventing migraine. These mechanisms might encompass enhancing GABAergic inhibition, blocking excitatory ion channels, and downregulating the expression of CGRP in brain tissue. These multifactorial actions collectively could underpin the migraine preventive effects of these antiepileptic drugs.

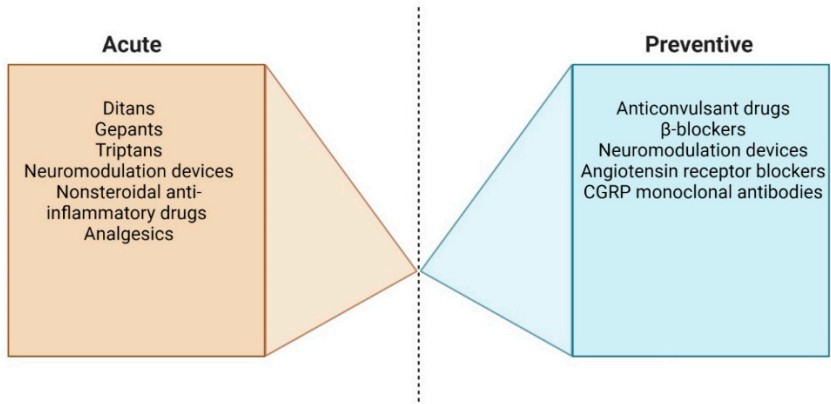

**Figure 3.** Migraine management.

6.4.2. Emerging Therapeutic Approaches in Migraine Management

1.  Introduction to novel medications and treatment strategies and Potential benefits and challenges associated with these approaches

    Goals of present research

    The goals of present research in migraine treatment are manifold and multifaceted, particularly in the context of elucidating and exploring the role of peptides or their re-

spective receptors that emerge during a migraine episode. Simultaneously, there is an unflagging pursuit of methods to obstruct the activation of the trigeminovascular system and the receptors of the neurotransmitters, which are intricately linked to the cascade of events leading to a migraine. These avenues of exploration are deemed paramount for the inception of innovative, effective, and targeted pharmacological interventions. These treatments are intended to serve both as acute therapeutic options and preventive strategies against migraine.

In light of the challenges and limitations associated with the use of triptans, a class of drugs conventionally employed in the treatment of migraine, there is a concerted effort within research and development programs to discover and develop new acute treatments. The aim here is to ensure these novel treatments are as effective as, if not more so, than triptans, whilst also being better tolerated by patients; furthermore, these treatments should ideally possess a distinctive, migraine-specific neural mechanism of action. An ideal feature of these new treatments would be the ability to avoid manipulating the vascular tone, thereby reducing potential side effects associated with vascular modulation [167].

In pursuit of these objectives, two primary biochemical pathways have been subjected to rigorous scrutiny—the calcitonin gene-related peptide (CGRP) pathway and the serotonin pathway. This intensive research has yielded two promising new classes of drugs, known as the gepants and the ditans, respectively. Both of these drug families are targeted interventions designed to alleviate the symptoms of migraine. In addition to their potential as acute treatments, the studies focusing on the CGRP pathway have also unearthed potential preventative strategies. This has culminated in the development of anti-CGRP monoclonal antibodies, which are being recognized as potential prophylactic treatments for migraine.

These promising new treatments are currently in an advanced stage of development, undergoing rigorous clinical trials. Some of these therapies are anticipated to be launched into the market in the near future, ushering in a new era of migraine management. This continued research endeavors to deliver more personalized and effective therapeutic options, minimizing the distress and impairment that migraine bring into the lives of those afflicted by them [167].

### 6.5. Competitive Environment

### 6.5.1. The Role of Calcitonin Gene-Related Peptide in Migraine

Regarded as one of the most potent vasodilators known, the calcitonin gene-related peptide (CGRP) exists in the following two forms within the human body: the alpha-CGRP, a 37-amino acid peptide that is predominantly expressed in primary sensory neurons of the dorsal root ganglia, trigeminal ganglia, and vagal ganglia; and the beta-CGRP, which is found primarily within intrinsic enteric neurons. The ubiquity of CGRP is seen in its widespread distribution across the cerebral and cerebellar cortex, thalamus, hypothalamus, inhibitory nociceptive nuclei of the brainstem, trigemino-cervical complex, and the trigeminovascular system [168].

Within the trigeminal ganglia, CGRP is found in cells that generate thinly myelinated A-delta fibers as well as unmyelinated C-fibers. Receptors for CGRP are present within the cortical and subcortical structures mentioned earlier. On trigeminal fibers, these receptors operate as autoreceptors, thereby governing CGRP release. Elevated levels of CGRP are detected during migraine attacks, although some research suggests conflicting evidence [167].

Notably, an intravenous infusion of CGRP was found to induce migraine attacks in about 60% of the patients studied. Interestingly, patients diagnosed with familial hemiplegic migraine, a rare type of migraine accompanied by aura, showed no sensitivity to CGRP. This could potentially be attributed to alterations in the levels of CGRP within their trigeminal system [169].

Experimental activation of trigeminal ganglion cells has been found to cause the release of CGRP. This release is inhibited in a dose-dependent manner by 5-HT1B/D

agonists, underscoring the importance of the trigeminal system as a potential target for CGRP receptor antagonists and triptans. CGRP, in addition to its vascular effects, has emerged as a key regulator of neuronal function, significantly influencing neurotransmitter systems like the glutamatergic system [170].

Drawing from these findings, drugs that modulate CGRP activity have shown promise in the future treatment of migraine. These include CGRP receptor antagonists, which compete with the body's naturally occurring CGRP at receptor binding sites and have been shown to be effective in the treatment of acute migraine attacks. Other approaches to modulate CGRP activity, such as the development of monoclonal antibodies against CGRP and the CGRP receptor, have been introduced recently [167].

### 6.5.2. CGRP Receptor Antagonists (The Gepants)

CGRP receptor antagonists, referred to as gepants, are small compounds that vie with the body's endogenous CGRP for receptor binding sites. The ability of these CGRP receptor antagonists to cross the blood–brain barrier remains uncertain. Despite their promise, the development journey of new emerging CGRP antagonists has been beset by challenges. Initial antagonists, such as olcegepant (BIBN4096BS), telcagepant (MK-0974), and MK-3207, demonstrated efficacy as acute treatments for migraine. However, they were burdened by unfavorable safety profiles.

In an initial proof-of-concept study in acute migraine treatment, the intravenously administered olcegepant at a dosage of 2.5 mg significantly outperformed placebo at a 2-h response rate, marking a potential breakthrough in acute migraine treatment [171]. Following this, telcagepant, an orally administered CGRP receptor antagonist, underwent a phase II proof-of-concept study that showcased the efficacy of a 300–600 mg dose. However, when telcagepant was tried as a daily preventive migraine treatment, it led to liver enzyme derangement, leading to the discontinuation of trials [172]. MK-3207, the third oral CGRP receptor antagonist developed and tested in migraine, showed superiority to placebo above the dose of 10 mg in 2-h pain freedom, but was discontinued due to liver toxicity issues.

A promising CGRP receptor antagonist, BI 44370 TA, was utilized in a phase II study to evaluate its safety, tolerability, and efficacy in the treatment of an acute migraine attack in episodic migraine sufferers [173]; however, studies on this agent have been discontinued as well.

### 6.5.3. Ubrogepant (MK-1602)

Ubrogepant (MK-1602) is a novel oral CGRP receptor antagonist that is chemically distinct from both telcagepant and MK-3207. The safety and efficacy of ubrogepant at varying doses was evaluated in a Phase IIb, multicenter, randomized, double-blind, placebo-controlled trial. Results showed a positive trend across all doses of ubrogepant for the 2-h pain freedom endpoint. Importantly, there were no observed post-treatment elevations of ALT > 3 ULN hand and no other abnormal laboratory values of clinical relevance, as found with the earlier CGRP antagonists. The success of this study led to the initiation of phase 3 clinical trials. Positive preliminary efficacy and safety results of two phase III multicenter randomized, double-blind, placebo-controlled clinical trials comparing ubrogepant 50 mg and 100 mg versus placebo (Achieve 1) and ubrogepant 25 mg and 50 mg versus placebo (Achieve 2) were recently presented at the American Headache Society (AHS) conference [174]. Despite these promising findings, further data on consistency of effect and safety in patients for whom triptans are contraindicated are needed to solidify its role as a viable alternative to triptans.

### 6.5.4. Rimegepant (BMS-927711): Overview and Clinical Trials

Rimegepant is a pioneering and unique calcitonin gene-related peptide (CGRP) receptor antagonist that bears a distinct chemical structure from telcagepant. In the sphere of migraine treatment, rimegepant's effectiveness and safety have been evaluated in a

rigorous phase II clinical trial. This trial was double-blind, randomized, placebo-controlled, and dose-ranging, involving a total of 885 participants.

The participants of the study were allocated randomly to receive one of six doses of BMS-927711 (10 mg, 25 mg, 75 mg, 150 mg, 300 mg, or 600 mg), sumatriptan (100 mg), or a placebo for the treatment of moderate to severe migraine attacks. The study was designed with the primary endpoint of achieving pain freedom two hours after the dose. Secondary endpoints were more comprehensive, including an endpoint consisting of the absence of headache pain, as well as a lack of symptoms, such as photophobia, phonophobia, and nausea, at two hours post-dose.

Along with these endpoints, the trial studied several other secondary efficacy and safety measures. Interestingly, a higher proportion of participants who received rimegepant 150 mg achieved the primary endpoint of being pain-free at two hours, amounting to 32.9%, which was significantly higher than the percentage observed for other doses of rimegepant ($p < 0.001$). For instance, the respective proportions were 31.4% for the 75 mg dose, 29.7% for the 300 mg dose, and 15.3% for the placebo group. Sumatriptan 100 mg proved superior to all doses of rimegepant, with a success rate of 35%.

With regards to the secondary efficacy endpoint of total migraine freedom, the dose of rimegepant 75 mg was the most effective, with a success rate of 28.2%. This dose was statistically superior to the placebo. However, sumatriptan 100 mg outperformed each dose of rimegepant at this secondary endpoint. The trial also reported that the proportion of patients who were headache-free for up to 24 h after dosing was higher for several doses of rimegepant and for sumatriptan compared to the placebo group.

In terms of safety, most adverse events (AEs) were mild to moderate, and none of the patients had to discontinue due to AEs. Two patients experienced increased hepatic enzymes reported as an adverse event, one in the rimegepant group and the other in the placebo group.

The results of this trial suggest that rimegepant's effectiveness is similar to sumatriptan 100 mg in treating migraine attacks, but with potentially fewer triptan-related side effects, such as paresthesia and chest discomfort. A phase III trial comparing the efficacy of rimegepant 75 mg with a placebo has recently been concluded. Moreover, an ongoing prospective multicenter open-label long-term safety study is expected to complete recruitment by late 2019. These two studies will contribute to understanding the consistency and safety of rimegepant in migraine therapy [160].

6.5.5. Atogepant (AGN-241689): The Future of Migraine Prevention

Atogepant, a small molecule with a distinct structure similar to that of ubrogepant, is the only CGRP receptor antagonist currently under investigation for migraine prevention. Its higher potency and longer half-life compared to ubrogepant make it suitable for preventive treatment.

The safety, efficacy, and tolerability of atogepant were evaluated in a phase II/III multicenter, randomized, double-blind, placebo-controlled, parallel-group study (NCT02848326). In this trial, adult patients were randomized to receive placebo, 10 mg QD, 30 mg QD, 30 mg BID, 60 mg QD, and 60 mg BID, respectively, and were treated for 12 weeks for the prevention of episodic migraine.

The primary efficacy endpoint was the change from baseline in mean monthly migraine/probable migraine headache days across the 12-week treatment period. All active treatment groups showed a statistically significant reduction from baseline in the primary efficacy parameter. In terms of safety, atogepant was well tolerated, with the most common adverse events being nausea, fatigue, constipation, nasopharyngitis, and urinary tract infection. The liver safety profile for atogepant was similar to placebo, with no indications of hepatotoxicity with daily administration over 12 weeks. The development program of this treatment is set to advance to the next stage [167].

6.5.6. Anti-CGRP and Anti-CGRP Receptor Monoclonal Antibodies

The CGRP pathway has also been targeted through the development of antibodies against CGRP and the CGRP receptor. This marks the first use of engineered antibodies in the field of migraine, and their development was initially met with some skepticism.

These monoclonal antibodies (mAbs), including galcanezumab (LY2951742), a fully humanized mAb anti-CGRP, fremanezumab (TEV-48125), a fully humanized mAb anti-CGRP, and eptinezumab, a genetically engineered humanized anti-CGRP antibody, target the ligand to prevent the binding of CGRP to its receptor. Erenumab (AMG 334) is a fully humanized mAb that targets the CGRP receptor.

These compounds have a favorable pharmacological profile that includes a long half-life, the absence of a vasoconstrictive effect or other significant hemodynamic changes [80]. Because of their high molecular weight, they do not cross the blood–brain barrier, suggesting a reduced likelihood of central nervous system-related side effects, which are commonly observed with current pharmacological prophylaxis treatments used in migraine.

Further enhancing their potential for long-term patient compliance, these mAbs can be administered either subcutaneously (sc) or intravenously (IV) at different rates ranging from once every three months to twice a month, depending on the compound.

In a series of methodologically similar randomized, double-blind, placebo-controlled Phase II and III clinical trials, the efficacy and safety of these novel treatments in the prevention of episodic and chronic migraine (CM) were explored [167].

6.5.7. Erenumab (AMG334)

Erenumab, marketed as Aimovig, is a novel migraine prophylactic belonging to the monoclonal antibody class of drugs. It operates by specifically targeting and blocking the calcitonin gene-related peptide (CGRP) receptor, a crucial element in the neurochemical pathway believed to play a role in migraine pathogenesis. The initial efficacy of erenumab was established through a phase II trial where participants suffering from episodic migraine received monthly doses of 70 mg over three months. The study outcomes exhibited a promising reduction in the number of monthly migraine days by 3.4 days compared to placebo [167].

This was further substantiated in the STRIVE study, a multicenter, phase III trial, which compared two doses of erenumab (70 mg and 140 mg) with a placebo over six months. The primary objective was to assess the change in the mean number of migraine days per month, while secondary objectives were to evaluate reductions in the severity of migraine and their impacts on physical function and daily activities. Both doses showed statistically significant reductions in the number of migraine days and severity compared to placebo. Notably, patients' disability scores, measuring the impact of migraine on daily activities, were also significantly improved [175].

A similar study, the ARISE trial, evaluated only the 70 mg dose of erenumab and confirmed its superior efficacy compared to placebo; however, unlike the STRIVE study, it did not reveal a significant improvement in the migraine disability scores. In both trials, erenumab demonstrated an excellent safety and tolerability profile with common side effects being minor, such as upper respiratory tract infection, injection site pain, and nasopharyngitis.

The long-term safety and efficacy of erenumab were further evaluated in a 5-year open-label extension of the phase II clinical trial. Here, participants with a history of inadequate response to up to two previous preventive treatments received erenumab 70 mg. This study demonstrated a reduction of 5 migraine days on average from an initial baseline of 8.8 migraine days per month. In addition, significant improvements were observed in disability and quality of life scores, indicating the long-term benefit of erenumab treatment [176].

Erenumab's performance was also assessed in chronic migraine patients in a randomized, double-blind, placebo-controlled phase II clinical trial. The participants received subcutaneous injections of either placebo, erenumab 70 mg, or erenumab 140 mg every

4 weeks for 12 weeks. Erenumab significantly reduced the number of monthly migraine days and monthly acute migraine treatments compared to placebo, thereby validating its preventive role in chronic migraine management [177]. As a result of these studies, erenumab was granted FDA approval in May 2018 for the prevention of migraine in adults.

6.5.8. Galcanezumab (LY2951742)

Like erenumab, galcanezumab is a humanized monoclonal antibody but differs in its mechanism. It inhibits the activity of CGRP by binding to the ligand itself rather than the receptor. The phase II proof-of-concept trials conducted in episodic migraine patients showed a mean reduction of 4.2 monthly migraine days with galcanezumab (150 mg) compared to a reduction of 3.0 days in the placebo group. The most commonly reported adverse events were erythema, upper respiratory tract infections, and abdominal pain [178].

A subsequent phase IIb clinical trial assessed the superiority of galcanezumab at varying doses (5, 50, 120, 300 mg) administered subcutaneously monthly for three months compared to placebo. The primary outcome was the mean change in migraine days from week 9 to 12 post-randomization. The 120 mg dosage significantly reduced migraine headache days compared with placebo [179].

Two phase III trials, EVOLVE-1 and EVOLVE-2, confirmed the efficacy of galcanezumab in reducing migraine days and improving the quality of life. Participants received monthly subcutaneous injections of galcanezumab at doses of 120 and 240 mg versus placebo for 6 months. Both studies met the primary and secondary efficacy endpoints at 6 months for both doses. The REGAIN study (NCT02614261), a 3-month double-blind study with a 9-month open-label extension for preventing migraine in chronic migraine patients, echoed the previous findings. The study met the primary endpoint at 3 months, showing that galcanezumab was significantly more effective in reducing monthly migraine days than placebo. Notably, it also showed a higher incidence of injection site reactions, erythema, and sinusitis in the galcanezumab groups [180]. These results highlight the promising role of galcanezumab in managing migraine, with long-term safety and efficacy data eagerly awaited.

6.5.9. Examination of Fremanezumab (TEV48125)

Fremanezumab, a fully humanized monoclonal antibody that targets the calcitonin gene-related peptide (CGRP), has been investigated for its potential use in the prevention of migraine. Its development was initially spurred by promising results from phase I studies, echoing the progress of other monoclonal antibodies against CGRP. To evaluate its efficacy and safety, a phase IIb study was carried out. The experimental protocol included a multicenter, randomized, double-blind, placebo-controlled design where participants were assigned to receive either 225 mg or 675 mg of subcutaneous (sc) TEV-48125 or a placebo. This was to be administered every 28 days for three months.

The primary objective of the study was to observe the mean decrease from the baseline in the number of migraine days during the third treatment cycle (weeks 9–12). In addition to this, safety parameters were also evaluated. In post-hoc analyses, investigators assessed the percentage of participants who achieved at least a 50% and 75% decrease in the number of migraine days relative to baseline. Both dosage levels of TEV48125 were found to meet the primary efficacy outcome, and no issues regarding safety or tolerability were identified. The most common adverse events associated with the treatment were mild pain or erythema at the injection site.

Following this, a phase III trial was conducted to further investigate the preventative effects of TEV-48125 (fremanezumab) in episodic migraine. This study was a randomized, double-blind, placebo-controlled, parallel-group trial that tested monthly sc fremanezumab injections of 225 mg or 675 mg following a quarterly dose regimen, against a placebo. The primary outcome was the mean change in the number of monthly migraine days per month over a 12-week period. Secondary efficacy endpoints included the proportion of patients

achieving at least a 50% reduction in the mean number of monthly migraine days from baseline to week 12, along with changes in migraine-related disability scores [181].

Both the monthly and quarterly regimens of fremanezumab met the primary efficacy endpoint, showing superiority over the placebo in reducing mean migraine days. No significant difference was found between the two fremanezumab regimens. Adverse events were most commonly injection site reactions, but the proportion of participants who discontinued due to these events was small (2%). This study also shed light on the possibility of using a single dose therapy given quarterly for migraine prevention, which is significant as the results were similar to those from monthly injections. This may open up new potential for multi-injection regimens in migraine prevention [182].

TEV-48125 (fremanezumab) was also tested for chronic migraine (CM) in a multicenter, randomized, double-blind, placebo-controlled, phase IIb study. Here, TEV-48125 was administered at different doses from those used in the episodic migraine trials as follows: 675 mg in the first treatment cycle and 225 mg in the second and third treatment cycles, or 900 mg monthly for three months, against a placebo. The efficacy endpoints for this trial varied from those of the episodic migraine studies. The primary outcome was the change from baseline in the number of headache-hours during the third treatment cycle (weeks 9–12), while the secondary endpoint was the change in the number of moderate or severe headache days. Both doses showed a significant reduction in the number of headache-hours compared to placebo and a significantly greater reduction in mean number of headache days [183].

After the promising results of the phase II study, a randomized, double-blind, placebo-controlled, parallel-group trial was conducted to further confirm the efficacy of fremanezumab for CM prevention. Participants with CM were randomized to receive fremanezumab either quarterly (a single dose of 675 mg at baseline and placebo at weeks 4 and 8) or monthly (675 mg at baseline and 225 mg at weeks 4 and 8) or a placebo. The primary endpoint was the mean change from baseline in the average number of headache days, as defined by the International Headache Society (IHS). Both doses met the primary endpoint and showed a significantly greater percentage of participants obtaining at least a 50% reduction in headache days compared to placebo.

Fremanezumab was found to be associated with a higher incidence of injection-site reactions than placebo, but the severity of these reactions did not significantly differ among the trial arms. Following these studies, fremanezumab (Ajovy) received FDA approval on 14 September 2018, making it the second anti-CGRP monoclonal antibody approved for preventing migraine in adults. Notably, it became the first drug of its kind to offer both quarterly and monthly dosing options [184].

### 6.5.10. Eptinezumab (ALD403)

Eptinezumab (ALD403) is an innovative product of genetic engineering, specifically, a humanized antibody that is designed to target both isoforms of human CGRP. CGRP, a molecule implicated in the pathophysiology of migraine, serves as a significant focal point for novel migraine therapeutics, including eptinezumab (see Figure 4) [65].

This novel therapeutic agent has been put through a phase II proof-of-concept study to evaluate its potential in managing episodic migraine. This study was designed with the primary objective of ascertaining the safety of eptinezumab following the intravenous administration of a single 1000 mg dose. As secondary objectives, the researchers investigated efficacy outcomes and gauged the extent of disability induced by migraine at the 12-week mark following infusion. They were particularly interested in discerning changes in the frequency of migraine days from the baseline to weeks 5–8.

Participants in the trial experienced an average of 8.4–8.8 migraine days per month at baseline, and adverse events were reported by 57% of eptinezumab recipients, compared to 52% in the placebo group. The adverse events included conditions like upper respiratory tract infection, urinary tract infection, fatigue, back pain, nausea, vomiting, and arthralgia,

most of which were of mild to moderate severity. Importantly, none of the serious adverse events reported were linked to the study drug.

When it came to efficacy, eptinezumab outperformed the placebo by achieving a statistically significant reduction in the mean number of migraine days from the baseline to weeks 5–8. The results also underscored a significant finding—a large proportion of participants receiving eptinezumab achieved a 50% reduction in migraine days at weeks 5–8. Interestingly, this trial observed a high placebo response rate, which might have been influenced by the intravenous administration of the drug, a departure from the delivery methods used in previous trials with anti-CGRP monoclonal antibodies.

Motivated by these promising findings, a phase III study was designed. Titled PROMISE 1, this randomized, double-blind, placebo-controlled trial sought to evaluate the efficacy and safety of various doses of eptinezumab in participants suffering from episodic migraine. The primary endpoint of this study was to identify changes in the mean frequency of migraine days over weeks 1–12, relative to a 28-day baseline period. In this study, participants were randomized to receive one of three doses of eptinezumab (300 mg, 100 mg, or 30 mg) or a placebo. The medication was administered via intravenous infusion every 12 weeks.

In the PROMISE 1 study, all doses of eptinezumab met the primary efficacy endpoint, demonstrating a significantly greater reduction in migraine days compared to the placebo group. Further, a significantly greater proportion of participants administered with eptinezumab experienced a 50% reduction in migraine days, providing more evidence for the drug's efficacy. The study did not find any significant safety issues, further supporting the suitability of eptinezumab as a therapeutic option.

Several studies with anti-CGRP monoclonal antibodies, including eptinezumab, have highlighted a remarkably rapid response to the active drug, usually noticeable within the first month following administration. Eptinezumab, in particular, demonstrated the ability to reduce migraine from the very first day post-administration and maintained similar improvement levels at 4- and 12-weeks post-infusion.

The evaluation of eptinezumab's efficacy, safety, and tolerability continued with phase II and phase III trials in patients with chronic migraine (CM). In a randomized, double-blind, placebo-controlled phase II study, different doses of eptinezumab were administered to test their effects against a placebo. Unique to this study, the primary endpoint was the percentage of patients achieving a 75% reduction in migraine days per month from baseline to week 12. Eptinezumab doses of 300 mg and 100 mg significantly outperformed the placebo group in achieving this endpoint.

These results paved the way for a phase III trial, known as PROMISE 2, to further investigate the safety and efficacy of eptinezumab for the prevention of chronic migraine. In this study, patients received either eptinezumab (300 mg or 100 mg) or a placebo, administered via infusion every 12 weeks. This study found that both doses of eptinezumab outperformed the placebo in reducing the mean number of monthly migraine days during the 12-week, double-blind treatment period. Notably, as early as day one post-infusion, a significant percentage of patients receiving eptinezumab demonstrated a reduction in migraine prevalence that was sustained through day 28.

Taken together, these studies affirm the potential of eptinezumab as a promising, efficacious, and well-tolerated therapeutic option for the management of episodic and chronic migraine. The drug's safety profile aligns well with that observed in previous studies, further bolstering its candidacy as an effective addition to the array of available migraine therapeutics [185].

**Figure 4.** The role of CGRP and the trigeminal system in migraine pathophysiology.

*6.6. Neuromodulation in Migraine*

6.6.1. Definition and Concept of Neuromodulation in Migraine Therapy

1.    Explanation of neuromodulation as a treatment modality

Neuromodulation is an innovative biomedical technique that manipulates or modulates the activities of the central or peripheral nervous system through the use of various stimuli such as electrical, magnetic, or chemical agents [186]. The technique hinges on the targeted delivery of these stimuli to specific neural sites within the body, with the express aim of altering nerve activity. Fundamentally, neuromodulation is non-destructive, reversible, and adaptable, highlighting its potential as a safe and effective approach for therapeutic interventions.

Recently, there has been an increased recognition of neuromodulation as a potentially superior treatment for migraine, compared to traditional medication therapies. The efficacy and safety of this technology are driving a paradigm shift in how both clinicians and patients approach the management of migraine. Instead of relying solely on pharmacological interventions, there is a growing interest in non-invasive neuromodulation therapies.

Among these therapies, non-invasive vagus nerve stimulation (nVNS) and single-pulse transcranial magnetic stimulation have gained notable attention due to their demonstrated effectiveness and safety profiles. These methods represent significant advancements in neuromodulation and are changing the landscape of migraine management [187].

In the modern healthcare context, neuromodulation technology holds substantial promise, particularly for vulnerable patient populations. For instance, expectant mothers, who need to avoid certain medications due to potential harm to the fetus, and patients who struggle with tolerating medications or find them ineffective, may greatly benefit from non-invasive neuromodulation therapies. The potential benefits extend not just to the realm of improved health outcomes, but also to the sphere of healthcare economics.

In specific circumstances, non-pharmacological neuromodulation techniques may prove to be a cost-effective alternative to traditional treatment modalities [188]. By reducing dependency on medications, these techniques can help circumvent the long-term costs

associated with drug therapy, such as costs related to side effects and long-term use. Furthermore, as these therapies improve in effectiveness and efficiency, they may help reduce the indirect costs of migraine, such as lost productivity and reduced quality of life.

In conclusion, neuromodulation represents a pioneering field in neuroscience that has the potential to revolutionize the treatment of migraine and other neurological disorders. Through the targeted use of electrical, magnetic, or chemical stimuli, this technique can modulate nerve activity in a way that is non-destructive, reversible, and adaptable. As research in this area continues to unfold, the healthcare industry will likely see an increasing shift toward these non-invasive, cost-effective, and patient-friendly treatment options.

2.    Overview of different types of neuromodulation techniques

Neuromodulation operates through the application of electrical or magnetic pulses to interact with or stimulate the central or peripheral pain pathways. This technique essentially targets the pain mechanisms in the human body with the aim of reducing the intensity of experienced pain. The application of electrical or magnetic stimuli can alter central neurotransmitters when they engage with pain circuits [189]. These modifications, in the context of treating acute migraine attacks, may potentially halt the processes that lead to the onset of an attack. For preventative purposes, these neuromodulatory changes aim to lessen the central sensitization that culminates in chronic headaches.

Traditional neuromodulation techniques engage the neurological system either centrally or peripherally through the skin, employing a changing magnetic field or an electric current to manipulate the mechanisms associated with headache-related pain. Both modes of delivery demonstrate rapid effectiveness, making them suitable for addressing acute symptoms; moreover, sustained application of these modalities might confer long-term preventative benefits [190]. Figure 5 offers a visualization of various neuromodulation techniques alongside their respective targets or sites of action.

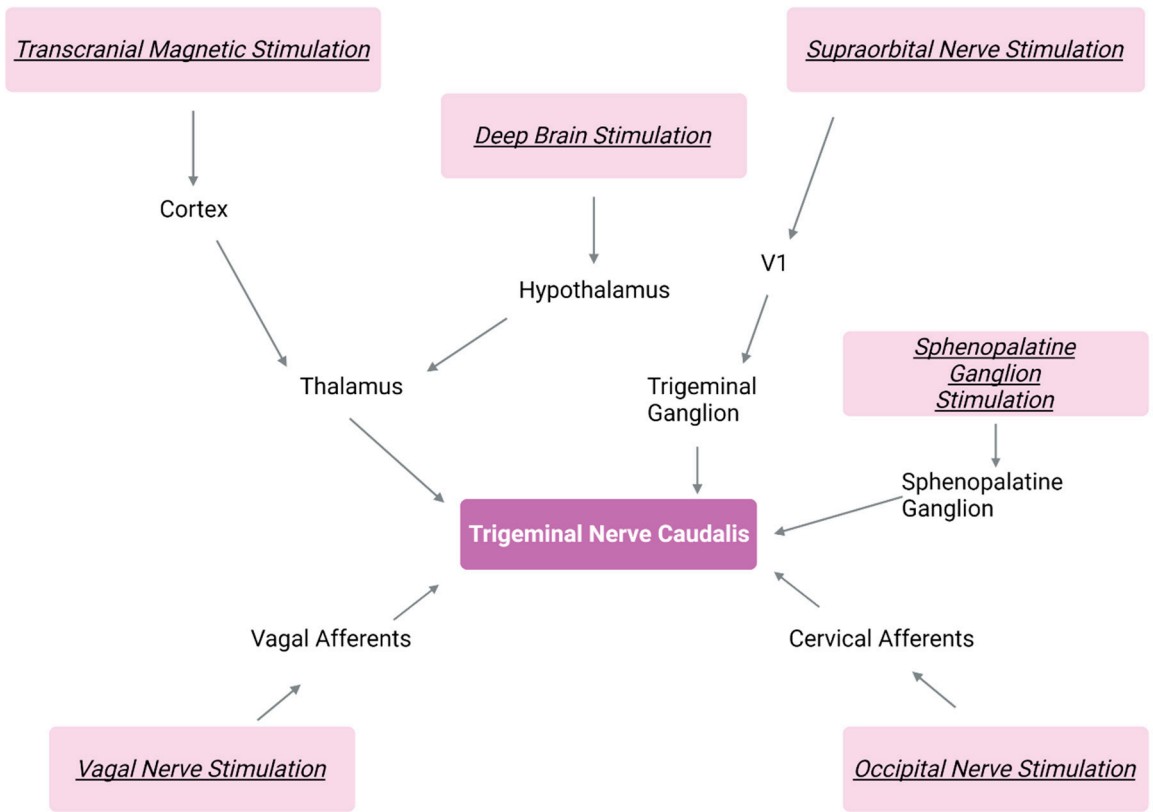

**Figure 5.** Several neuromodulation techniques and their respective site of action or targets.

Neuromodulation is implemented via a device that modifies brain cell activity utilizing electrical or magnetic stimulations. These devices exhibit diversity in their modes of operation. Some are designed to halt attacks, while others are utilized for preventative purposes. The commonality among them lies in the principle of altering the activity within nerve pathways. These devices, alternatively referred to as stimulators, can be categorized based on their operational parameters. Neuromodulation devices may employ magnetic, electrical, or temperature-changing stimuli. They may be invasive or non-invasive, and their design may range from portable, easy-to-use devices to those that necessitate surgical placement [187].

In the realm of neuromodulation, different techniques and devices offer a range of options to tailor treatments according to the patient's specific needs. As our understanding of the mechanisms behind migraine deepens, the applications of neuromodulation will continue to evolve, potentially offering more effective and personalized treatments for patients suffering from migraine and other neurological conditions.

3. Neuromodulation Modalities

With the emergence of neuromodulation and the recognition of its potential for preventative treatment in chronic pain conditions, such as migraine and cluster headaches, there has been significant development in the field of non-invasive neuromodulation techniques. These advancements offer alternative treatments that pose minimal risk to the patient while maximizing the potential for pain management and relief. Not only have several of these techniques successfully passed through clinical trial phases, but they have also found their way into the marketplace where they are actively being utilized for patient treatment.

Simultaneously, numerous other neuromodulation techniques are currently in various stages of clinical trials, showing the ongoing growth and exploration in this sector of medicine. These techniques, aimed at managing acute migraine attacks and chronic pain, are progressing towards market utilization, continually broadening the treatment options available for these conditions [191].

Figure 6 offers a visual representation of the different non-invasive neuromodulation techniques employed in the management of acute migraine episodes. This visual aid helps provide a better understanding of the variety and extent of non-invasive techniques available, each with its distinct advantages and specific use-cases, contributing to the expansive repertoire of neuromodulation methods.

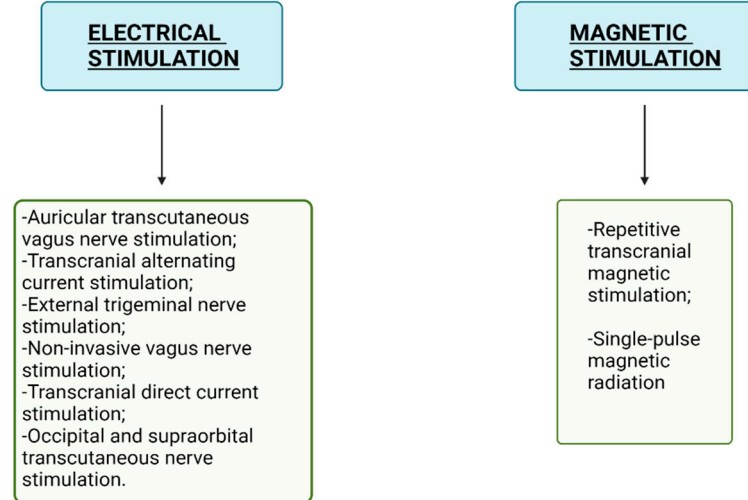

**Figure 6.** Different non-invasive neuromodulations in the management of acute attacks of migraine.

The evolution and diversity in neuromodulation techniques symbolize the adaptability and progressive nature of contemporary medicine. Through continuous research and development, these techniques are becoming increasingly refined, potentially providing

better, more targeted treatment for patients suffering from acute migraine and other chronic pain conditions. It signifies a remarkable shift in our approach to pain management, with techniques that prioritize patient safety and comfort while not compromising on effectiveness [187].

6.6.2. Effectiveness of Neuromodulation in Managing Migraine: Review of Clinical Studies on the Use of Neuromodulation for Migraine—Assessment of the Benefits and Limitations of Neuromodulation

1.    Vagus Nerve Stimulation

The gammaCore Sapphire™, crafted by electroCore, Inc. (Basking Ridge, NJ, USA), is a handheld non-invasive vagus nerve stimulator (nVNS) designed for transcutaneous application to the vagus nerve on either side of the neck. The FDA first granted clearance to this device in April 2017, initially for the treatment of acute pain associated with episodic cluster headaches in adults. Now, it has expanded FDA approval, covering both the acute and preventive treatment of migraine-related pain for individuals aged 12 or older.

Four major nVNS trials have been conducted concerning migraine, consisting of two studies for episodic migraine (EM), one tailored for chronic migraine (CM), and another one incorporating CM. The EVENT study, which was the first nVNS trial for CM, primarily sought to assess safety and tolerability in a pilot feasibility format. This randomized double-blind sham-controlled research focused on CM prevention. Though the efficacy endpoints (a change in monthly headache days after 2 months; $-1.4$ vs. $-0.2$, $p = 0.59$) did not show statistical significance, there were hints of potential benefits with prolonged use. After an 8-month treatment period, 15 subjects witnessed a mean change of $-7.9$ (95% CI $-11.9$ to $-3.8$, $p < 0.01$) [186].

Another noteworthy trial was the PREMIUM II study, a randomized sham-controlled double-blind study that included both CM and EM subjects. Unfortunately, it was cut short due to the COVID-19 pandemic. With a total of 300 participants enrolled and a modified intention-to-treat (mITT) subgroup of 113 subjects analyzed, the PREMIUM II study discovered a non-significant reduction in monthly migraine days (verum vs. sham: $-3.1$ vs. $-2.3$ headache days, $p = 0.233$) in the mITT cohort. However, 44.9% of verum-treated individuals experienced at least a 50% decrease in migraine days, in contrast to 26.8% in the sham group ($p = 0.048$) [187].

Clinical trials have demonstrated that the gammaCore Sapphire™ is safe and generally well-tolerated, with no significant treatment-linked adverse events. Some common side effects reported in the CM trials were facial pain, gastrointestinal symptoms, and upper respiratory tract infections. While the preliminary data indicate that nVNS may hold promise for treating CM, further exploration and evidence are needed to solidify these findings.

2.    Remote Electronic Neuromodulation

Nerivio® (Theranica Bio-Electronics Ltd., Montclair, NJ, USA) is a remote electronic neuromodulation (REN) device that has been FDA-approved for the acute treatment of migraine (both EM and CM) in patients aged 12 years and older. The device comprises an armband that emits electronic stimuli controlled by a smartphone app. It is recommended that patients begin use within 60 min of the onset of a migraine or aura. The stimulation lasts for 45 min and can be adjusted for intensity by the individual user via the app, which also features a migraine diary for logging headaches and usage sessions.

Following two randomized double-blind sham-controlled trials that led to the initial FDA clearance of Nerivio for use in EM in 2019, two subsequent open-label observational studies (TCH-005, TCH-006) contributed to the FDA clearance for CM in 2020. In the TCH-005 study, 42 subjects were enrolled, with 210 evaluable treatments carried out by 38 participants. The TCH-006 study evaluated a total of 493 evaluable treatments from 91 participants out of the 126 enrolled [192].

The device-related adverse events were generally related to topical peripheral sensations such as warmth, itching, arm pain, redness, and numbness.

3. Electrical Stimulation of the Trigeminal Nerve

The CEFALY DUAL device, a next-generation evolution of the original electrical trigeminal nerve stimulation (eTNS) unit designed by CEFALY-Technology (Seraing, Belgium), received the stamp of approval from the FDA for the acute as well as preventive management of migraine in adults. As of late 2020, this device is available over-the-counter. CEFALY DUAL offers two customizable settings—an ACUTE program, which utilizes a 100 Hz frequency over a 60 min session for instant relief, and a PREVENTIVE program, which operates at a lower frequency of 60 Hz for 20 min daily to thwart the onset of migraine [193,194].

The therapeutic potential of the CEFALY device for both acute and preventive treatment of migraine is supported by two randomized controlled trials, with one trial specifically mentioning CM [8]. The ACME study, a randomized double-blind sham-controlled trial involving patients with migraine (likely including but not specifying CM), revealed that using CEFALY during the onset of a headache provided more significant pain reduction compared to the sham group ($-3.46 \pm 2.32$ vs. $-1.78 \pm 1.89$; $p < 0.001$). The study likely comprised mostly patients with episodic migraine (EM), as those who had used Botox in the previous four months were excluded [193,194].

Three open-label observation studies, two involving CEFALY and one using supraorbital transcutaneous electrical nerve stimulation (TENS), have specifically targeted CM. In these studies, the CEFALY device was found to decrease monthly migraine days and acute medication consumption. The adverse events were minimal and reversible, with the most common being paresthesia (2.03%), changes in arousal (mostly fatigue, sometimes insomnia, 0.82%), headache (0.52%), and skin allergy to the electrode (0.09%) [13]. These findings suggest that eTNS could be beneficial for CM treatment [192].

4. Single-Pulse Transcranial Magnetic Stimulation

eNeura Inc.'s sTMS mini™, hailing from Baltimore, MD, USA, is an innovative device that administers single-pulse stimulation to the rear of the head. This stimulation method is believed to modify the excitability of the cerebral cortex by halting cortical spreading depolarization waves and curbing thalamocortical signaling [190]. Initially conceived for episodic migraine patients exhibiting aura, the device has secured FDA clearance for both the acute and prophylactic treatment of migraine in individuals aged 12 years and up. The preventive regimen necessitates a twice-daily treatment involving 4 pulses (a pair of consecutive pulses, a pause of 15 min, followed by another 2 pulses). For acute treatment, the procedure entails three successive pulses when a migraine begins, with an option to administer more pulses at quarter-hour intervals if required [191].

The FDA endorsement of this device came on the heels of a randomized, double-blind, sham-controlled trial. Following this, an open-label observational investigation, dubbed the ESPOUSE study, was conducted. This study saw participation from 13 (amounting to 10%) chronic migraine sufferers who used the device daily for both preventive measures (the aforementioned 4 pulses administered twice daily) and acute relief (3 pulses with the possibility of repetition every 15 min for two subsequent sessions). Although the specific effects of daily sTMS application on chronic migraine were not delineated in detail, the research indicated that sTMS effectively slashed monthly headache days, reduced the need for acute medication, and brought down the Headache Impact Test-6 (HIT-6) score. The treatment regimen involving sTMS was generally well-received by participants. The predominant side effects cataloged were sensations of lightheadedness, tingling, and the occurrence of tinnitus [191].

5. Combined Occipital and Trigeminal Nerve Stimulation

The Relivion® is a user-operated stimulation device that secured FDA clearance in the early months of 2021. It is adept at delivering electrical surges to six branches spanning both the occipital and trigeminal nerves. The device comes equipped with pre-set settings, allowing for six treatment cycles (providing uninterrupted stimulation for a 48-h window) as and when needed, primarily for delivering swift relief from migraine episodes. The green

light for its market entry was largely due to findings from a widespread study dubbed the RIME study. This investigation involved 131 episodic migraine sufferers who underwent stimulation for a one-hour period [187].

A look back at previous studies revealed that the concurrent stimulation of both occipital and trigeminal nerves, executed via an implanted mechanism, yielded positive results both in the immediate aftermath and over extended periods. Specifically, 75% (or 4 out of 16) of the subjects with stubborn CM observed short-term benefits, while half of the participants (8 out of 16) reported sustained relief [192]. These initial findings hint at the potential of Relivion® as a viable therapeutic avenue for CM. However, a more comprehensive dataset is required to substantiate these preliminary yet promising outcomes.

### 7. Investigational Devices

#### 7.1. Transcranial Direct Current Stimulation (tDCS)

Transcranial direct current stimulation (tDCS) employs a gentle current, typically ranging from 1 to 2 milliamperes, delivered through sponge electrodes arranged in a variety of montages. This technology has been investigated as a potential treatment for headache disorders. Despite the exact mechanism of action remaining unclear, it is believed that tDCS may influence network-level neural information processing without directly affecting neural spiking or membrane potential. This process could potentially modify brain connectivity, thereby enhancing placebo effects while reducing nocebo effects. However, it is important to note that many of the tDCS trials for migraine were pilot studies of low to moderate quality. These trials have employed varying stimulation duration, current ampere, polarity, montage, and the number of sessions, thus, a universally optimized stimulation protocol has not yet been established [192].

In recent years, spanning half a decade, four rigorously designed, sham-controlled trials have honed in on transcranial direct current stimulation (tDCS) as a possible treatment route for chronic migraine (CM). These studies, distinct in design, likely adhered to a single-blind methodology. Complementing these, two studies adopting an open-label approach were undertaken. Noteworthy findings were presented by Andrade and team, pinpointing that anodal stimulation, targeted either at the left primary motor area (M1) or the dorsolateral prefrontal cortex (DLPFC), as opposed to sham procedures, significantly diminished HIT-6 scores and alleviated pain intensity [193]. In a contrasting study spearheaded by Dalla Volta and colleagues, it was discovered that cathodal stimulation, specifically targeting the coolest forehead point, surpassed sham stimulation in terms of reducing monthly headache occurrences, the frequency of attacks, and their duration. Adding to this narrative, two niche open-label research initiatives underscored a notable reduction in headache episodes observed 30 days post anodal stimulation [194]. In stark contrast, findings documented by Cerrahoglu Sirin and team indicated no discernible difference in the monthly headache frequencies a month subsequent to either anodal or sham stimulation [195]. Further amplifying the narrative of inconclusive results, a comprehensive investigation by Grazzi et al. failed to delineate any significant differences when comparing anodal tDCS, cathodal tDCS, and sham stimulation at the 6 and 12-month mark. This was specifically observed in CM patients who were in the throes of abrupt medication withdrawal due to overuse [196].

Conclusively, the true efficacy of tDCS in offering a preventive strategy against CM, especially when gauged several months post-intervention, remains shrouded in ambiguity. For future tDCS trials to shed more clarity and establish therapeutic consistency, there is an imperative need to adopt a standardized protocol, meticulously outlining the polarity, montage, number of sessions, repetition intervals, and the definitive endpoints.

#### 7.2. Repetitive Transcranial Magnetic Stimulation (rTMS)

Repetitive transcranial magnetic stimulation (rTMS) devices deliver a sequence of rapid pulses. These pulses are engineered to generate a minute, focused cortical electric

current targeting specific brain areas like the M1 and DLPFC. Intriguingly, these areas are pivotal in modulating motor-thalamus-brainstem as well as prefrontal-thalamic-cingulate signaling pathways. As a general rule of thumb, high-frequency stimulations, ranging from 5 to 20 Hz, are believed to amplify cortical excitability, interact with diverse neurotransmitter and opioidergic networks, and mold neuronal plasticity [197,198]. The therapeutic potential of rTMS has been officially recognized by the FDA for the management of severe depression and obsessive-compulsive disorders. Even with a solid safety track record, the efficacy of rTMS in tackling pain disorders, post-traumatic headaches, and primary headache syndromes is currently under the microscope. Recent comprehensive reviews propose that high frequency rTMS, specifically targeting the motor cortex, might emerge as a promising migraine management strategy. However, the need of the hour is robust, high-quality randomized controlled trials (RCTs) with unified protocols to substantiate this therapeutic proposition [199,200].

In the recent half-decade, both open-label and randomized controlled trials have delved deep into the potential of rTMS for CM management [187]. Though certain open-label initiatives incorporated CM, they steered clear of presenting efficacy data and, hence, will not be the focus here [201]. An exploratory study helmed by Rapinesi et al. broadcasted a marked dip in migraine episodes, dependency on rescue medications, pain intensity, and scores measuring depression. This was observed after the combination of deep TMS, targeting the left DLPFC with a specific protocol, and conventional treatments [202]. A comparative study by Shehata et al., juxtaposing rTMS with onabotulinumtoxin A, revealed that rTMS, designed with a specific frequency and targeting the left M1, echoed the therapeutic efficacy of onabotulinumtoxin-A in managing CM; however, the rTMS effects waned noticeably after a span of eight weeks [203]. A research endeavor by Kalita et al., contrasting left M1 rTMS over a three-month period in CM and chronic tension-type headache patients, underscored a significant reduction in headache episodes post rTMS treatment within the group; however, when pitting the two groups against each other, the results were not statistically significant [204]. Granato et al., in their research with CM patients also grappling with medication overuse headache (MOH), could not identify any clear advantages of rTMS over its sham counterpart, especially when metrics like monthly headache days, symptomatic drug dependency, and Migraine Disability Assessment (MIDAS) scores were evaluated after a 120-day span [205]. An intriguing facet of this study was the sham stimulator's capacity to mimic the vibratory sensation of actual stimulation. Whether this mimicry carries any real therapeutic value remains an enigma.

*7.3. Occipital Nerve Stimulation (ONS)*

Occipital nerve stimulation (ONS) is a form of treatment that involves the use of an implantable device with electrodes positioned near the occipital nerves and a pulse generator in the chest. This type of treatment has been researched for many years for conditions like occipital neuralgia and refractory chronic migraine (CM). While the precise mechanism is not fully understood, it is believed that ONS may counterbalance trigeminally mediated central sensitization or restore the lost conditional pain modulation in these patients. A number of multi-center randomized sham-controlled trials, published over five years ago, showed improvements in areas such as headache frequency, intensity, and disability [192].

Within the last five years, open-label studies have been published on the use of ONS for chronic migraine [195–197]. However, the specific parameters of the stimulation and the study endpoints were quite varied between studies. Miller et al. analyzed a group of 53 intractable CM patients and discovered an 8.51-day reduction in monthly moderate-to-severe headache days after bilateral ONS electrodes implantation [195]. Similarly, Garcia-Ortega et al. studied 37 refractory CM patients and reported significant pain reduction [196]. However, it is worth noting that up to 20% of patients reported adverse events such as infection, lead migration, and stimulation-related symptoms one year after treatment [197].

Despite the potential side effects, ONS seems to be a promising device for chronic migraine; however, no ONS device has been cleared by the FDA for use in treating migraine yet.

*7.4. Spinal Cord Stimulation (SCS)*

Spinal cord stimulation (SCS) is another method that has been utilized to manage intractable headache. It involves placing SCS electrodes into the high-cervical epidural space (C2/3) with a pulse generator implanted subcutaneously. This technique has shown promising results, with patients experiencing a significant reduction in mean pain intensity and a decrease in the median number of migraine days. Nevertheless, issues such as infections and lead dislocations have been reported. Despite these concerns, the effectiveness of SCS for chronic migraine is still under investigation [198].

Another interesting approach being explored is the exposure to green light. This might help modulate nociception and anxiety. While non-green light stimuli seemed to exacerbate pain intensity during a migraine attack, exposure to green light was seen to reduce pain intensity in around 20% of the patients. In a small crossover study, daily green light exposure resulted in a significant reduction in the number of headaches in CM patients. Due to its potential effectiveness and safety, this approach warrants further investigation [192].

## 8. Clinical Perspective

There are presently almost half a dozen devices cleared by the U.S. Food and Drug Administration (FDA) specifically for treating migraine. Interestingly, the clearance for several of these devices has also been extended to cover adolescent patients aged 12 or older. However, it is crucial to note that not all of these FDA-cleared devices have been thoroughly investigated for use in chronic migraine (CM). The only randomized sham-controlled trial conducted for CM was undertaken utilizing the gammaCore device, and unfortunately, it failed to meet its primary efficacy endpoint.

Various open-label observational studies have employed devices, such as CEFALY or Nerivio, to assess pain reduction in patients with CM. However, these studies lack blinding and are subject to potential selection and reporting biases. Furthermore, certain studies have included both episodic migraine (EM) and CM patients, yet failed to provide a detailed breakdown of the number of CM cases or any response data specifically related to CM.

In addition, the absence of established trial guidelines that address the unique aspects and challenges associated with neuromodulation device trials for migraine has led to substantial variation in study endpoints, types of control, and the populations analyzed (intention-to-treat vs. per-protocol). As a result, it can be difficult to compare the outcomes of different studies. In an effort to address these issues, the International Headache Society has published recommendations for evaluating neuromodulation devices in both the acute and preventive treatment of migraine [199].

As the application of these devices becomes more common, we anticipate the emergence of larger, higher-quality studies that adhere to established clinical trial guidelines. These studies are essential to fully establish the benefits of these devices in the treatment of CM. Furthermore, the conduct of high-quality trials should also ideally encourage insurance companies to broaden their coverage to include more neuromodulation devices.

## 9. Discussion

Migraine remains an intriguing neurological disorder with complex mechanisms and pathophysiology that are still being researched extensively. Migraines are believed to be hereditary conditions characterized by increased responsiveness of cortical and subcortical networks; however, the triggers and factors contributing to its onset and progression remain unknown.

Migraine's multidimensionality can be seen through the following various phases: premonitory phase, headache pain, postdromal phase, and sometimes aura phase. Each of

these involves interactions among hypothalamus nuclei, cortical regions, and trigemino-vascular pathways, resulting in characteristic symptoms experienced during an attack.

Despite these challenges, migraine research has made significant advances, providing light on potential therapeutic targets and validating CGRP as an effective target for both acute and preventative treatments—giving hope of the better management of migraine episodes as well as improved quality of life for affected individuals.

However, much work remains to be performed in understanding its intricate processes and developing effective treatment and prevention measures. More awareness, additional funding for research, collaboration among scientists, healthcare providers, and patients, as well as increased collaboration, will all play an integral role in increasing understanding and alleviating migraine's burden.

As we discover more about migraine, it is crucial that we provide empathy, support, and personalized care to those living with this neurological challenge. By harnessing the collective knowledge and dedication of scientific communities worldwide, we can make significant advances in migraine research that ultimately benefit millions living with this neurological disorder [200].

**Author Contributions:** Conceptualization, H.P. and A.V.C.; methodology, T.-L.T.; soft-ware, I.-A.F.; validation, R.-A.C.-B., L.-A.G. and H.P.; formal analysis, D.-I.D.; investigation, A.A.P.; resources, A.B.; data curation, L.-A.G.; writing—original draft preparation, R.-A.C.-B. and A.V.C.; writing—review and editing, R.-A.C.-B., L.-A.G. and A.V.C.; visualization, A.B.; supervision, D.-I.D.; project administration, A.V.C.; funding acquisition, A.V.C. All authors have read and agreed to the published version of the manuscript.

**Funding:** This research received no external funding.

**Institutional Review Board Statement:** Not applicable.

**Informed Consent Statement:** No requirement for ethical approval.

**Data Availability Statement:** All Data is available on PubMed.

**Conflicts of Interest:** The authors declare no conflict of interest.

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
