# Peer review of "Migraine: Advances in the Pathogenesis and Treatment"

_2035-8377, doi:10.3390/neurolint15030067_

Round 1

Reviewer 1 Report

Well written review with many clear genetic, molecular, radiologic, and clinical explanations about chronic migraine.Tha only suggestion is to update Table 3 because eptinezumab and rimegepant (as prevention and migraine attacks treatment) are approved by FDA and EMA and now available.

Author Response

Dear reviewer,

I’ve updated Table 3 with the current approval status of each pharmacological substance

Thank you for your positive review and helpful recommendations!

Reviewer 2 Report

The paper presented to me for review is a very broad review on the pathogenesis and treatment of migraine.

Although the paper is very comprehensive and informative it contains a lot of inaccurate and erroneous data and needs considerable rewriting before it can be considered for publication:

1. The paper, in my opinion, is too extensive - it contains a great deal of information often written in very poor scientific language. The whole article gives the impression that it is written by several people "piece by piece" - fragments are uneven in content and language. Often information is repeated many times in several places.

2. The title should read differently. There is no such thing as migraines! Migraine is a singular number and a specific neurological disease within which we distinguish different forms. I would suggest simplifying the title to: Migraine: Advances in the Pathogenesis and Treatment because the term chronic headache disorders is also not appropriate for this entity.

3. The text repeatedly uses the term migraines - this should be changed, as well as migraineurs (pp. 11, 399) - it is a pejorative term for patients suffering from migraine.

4. The paper is based on OLD literature, most of the work is from before 2012 where the greatest progress is the last years!

5. The introduction lacks reference to the prevalence of migraine in children - we already have new papers on this issue that should be cited: PMID: 36782182, PMID: 35190383

6. Page 4 discusses biomarkers - I think it is a mistake to refer to additional tests as "techniques to assess the course of the disease" - the course of the disease has not been proven to correlate with changes in brain MRI and neurophysiological techniques are not useful at all in daily clinical practice - this may mislead clinicians who recommend EEG in migraine and there is no justification for this. This should be removed or clarified.

7. Genetic markers (page 7) - chapter based on outdated literature! there is a lack of recent articles on this topic that describe it accurately: PMID: 35735024 and PMID: 36800925

8. Page 13- Investigators... The entire chapter is based on only one and on top of that an old citation. New information about different therapeutic targets in migraine is missing: PMID: 37540009, PMID: 37370051, PMID: 37029836 and PMID: 37584847

9. Table 3 contains outdated information (developed stage) - many drugs have long been available to patients and in clinical trials e.g. rimegepant

10. Figure 3 - ergotamine is already abandoned in all recommendations and preventive treatment lists triptans which are not true and neurotoxins - what toxins?

11. Long sections of the paper are based on single citations.

12. References should be verified and refer to new studies.

Author Response

Dear reviewer,

I’ve modified the title according to your recommendation

Terms migraines and migraineurs are changed to migraine in the whole manuscript

New studies were added, with more relevant information, especially those outstanding papers recommended

Data for prevalence of migraine in children is now included in our study. Moreover, a subchapter was added regarding medical treatment possibilities of migraine in pediatric populations

I’ve added a new paragraph with clarifications regarding efficiency and relevance of biomarkers for clinical practice in accordance with your recommendation

I’ve added newer genetic markers, thank you for the helpful studies listed! Transgenic mice studies were included

I’ve updated Table 3 with the actual status of approval for each pharmacological substance

I’ve updated Figure 3 according to your indications

More citations from the last years were added

Thank you for your significant helpful review and for the relevant studies recommended

Round 2

Reviewer 2 Report

The authors have responded fully to my comments. The manuscript has been revised in accordance with my suggestions. The manuscript is currently ready for printing.